# General recipe to realize photonic-crystal surface-emitting lasers with 100-W-to-1-kW single-mode operation

Takuya Inoue [1,3✉], Masahiro Yoshida [2,3], John Gelleta [2,3], Koki Izumi[2], Keisuke Yoshida[2], Kenji Ishizaki[2], Menaka De Zoysa[1] & Susumu Noda [1,2✉]

Realization of one-chip, ultra-large-area, coherent semiconductor lasers has been one of the ultimate goals of laser physics and photonics for decades. Surface-emitting lasers with two-dimensional photonic crystal resonators, referred to as photonic-crystal surface-emitting lasers (PCSELs), are expected to show promise for this purpose. However, neither the general conditions nor the concrete photonic crystal structures to realize 100-W-to-1-kW-class single-mode operation in PCSELs have yet to be clarified. Here, we analytically derive the general conditions for ultra-large-area (3~10 mm) single-mode operation in PCSELs. By considering not only the Hermitian but also the non-Hermitian optical couplings inside PCSELs, we mathematically derive the complex eigenfrequencies of the four photonic bands around the $\Gamma$ point as well as the radiation constant difference between the fundamental and higher-order modes in a finite-size device. We then reveal concrete photonic crystal structures which allow the control of both Hermitian and non-Hermitian coupling coefficients to achieve 100-W-to-1-kW-class single-mode lasing.

[1] Photonics and Electronics Science and Engineering Center, Kyoto University, Kyoto 615-8510, Japan. [2] Department of Electronic Science and Engineering, Kyoto University, Kyoto 615-8510, Japan. [3] These authors contributed equally: Takuya Inoue, Masahiro Yoshida, John Gelleta. ✉email: t_inoue@qoe.kuee. kyoto-u.ac.jp; snoda@kuee.kyoto-u.ac.jp

Realization of one-chip single-mode high-power semiconductor lasers, which surpass the performance of all other lasers such as solid-state lasers, fiber lasers, and gas lasers, has been one of the ultimate goals of laser physics and photonics for decades. The demand for such semiconductor lasers has been rapidly increasing for a wide variety of applications including next-generation laser processing[1,2] and remote sensing[3,4]. Conventional semiconductor lasers such as edge-emitting lasers and vertical-cavity surface-emitting lasers involve fundamental difficulties for single-mode high-power operation because an increase of the device size inevitably results in the onset of multiple transverse-mode lasing[5–8]. On the other hand, photonic-crystal surface-emitting lasers (PCSELs)[9–16], which utilize a two-dimensional standing-wave resonance at a singularity point (Γ point, etc.) of the photonic band for lasing oscillation, show promise for overcoming this difficulty; the mutual coupling coefficients among propagating waves and radiative waves inside PCSELs can be manipulated by the unit cell design[11,14], which can greatly enhance the threshold margin between the fundamental mode and the other higher-order modes. Towards the realization of ultra-large-area single-mode PCSELs, a double-lattice photonic crystal, in which two lattice-point groups are shifted in the $x$ and $y$ directions by one quarter of the lattice constant $a$ (which is almost equal to the wavelength in the material), was recently proposed[14]. In the double-lattice photonic crystal, the optical feedback (coupling coefficient) is weakened by the destructive interference for 180° diffraction of the waves reflected by each lattice point, with which the optical losses of the higher-order modes are increased more than those of the fundamental mode. Based on this concept, 10-W-to-20-W-class single-mode lasing was experimentally demonstrated with PCSELs with a diameter of as large as 400–500 μm[15,16]. In addition, by introducing destructive interference of not only 180° diffraction but also 90° diffraction, the possibility of single-mode lasing in a larger device area with a diameter up to 2.5 mm was suggested[14]. However, the conditions to realize single-mode lasing in an even larger area (3–10 mm), which are essential to realize 100-W-to-1-kW single-mode lasing, have not yet been derived, nor have the concrete photonic crystal structures to satisfy the conditions been clarified. This is because the non-Hermitian process inside photonic crystals, which accompanies energy loss, has not been so far utilized effectively.

In this paper, we first mathematically derive the complex eigenfrequencies of the four photonic bands around the Γ point, which form two-dimensional broad-area cavity modes, by considering not only the Hermitian but also the non-Hermitian optical couplings inside PCSELs using the three-dimensional coupled-wave theory[17,18]. Next, we provide a formula for the threshold margin of the fundamental mode over the higher-order modes in the finite-size device, and reveal the general conditions for the Hermitian and non-Hermitian optical coupling coefficients to realize broad-area single-mode lasing. We show that the key to realize such broad-area single-mode lasing is in the utilization of a carefully designed double-lattice photonic crystal structure with an appropriate backside reflection, by which the flexible control of both the Hermitian and non-Hermitian coupling coefficients becomes possible. Finally, we investigate the lasing stability of the designed PCSELs through a comprehensive analysis by considering carrier-photon interactions, and reveal more detailed requirements for the Hermitian and non-Hermitian coupling coefficients, with which stable 100-W-to-1-kW-class single-mode lasing can be expected in an ultra-large lasing diameter (3–10 mm).

## Results

**Hermitian and non-Hermitian optical couplings inside PCSELs.** Figure 1 shows a schematic of mutual couplings of waves inside a square-lattice PCSEL, where a double-lattice photonic crystal is employed. The Hermitian couplings, which express the optical couplings without accompanying energy loss (or vertical radiation loss), are shown in Fig. 1a, while the non-Hermitian couplings, which express the optical couplings with accompanying energy loss (or vertical radiation loss), are shown in Fig. 1b. Here, a back-side reflector is placed beneath the photonic crystal to reflect the downward radiation to the upward direction and to control especially the magnitude of non-Hermitian coupling coefficient. In the analysis below, we focus on the transverse-electric-like (TE-like) modes of the photonic crystal since the active layer of the typical PCSELs consists of multiple quantum wells with TE gain. Note that the analysis below can be also applied to other square-lattice photonic crystals that have the same reflection symmetry as the double-lattice photonic crystal (i.e. reflection symmetry along $y = x$).

According to Bloch's theorem, the electric fields distribution $\mathbf{E}(\mathbf{r})$ inside the photonic crystal with a lattice constant of $a$ is expressed by the superposition of many propagating plane waves as follows;

$$\mathbf{E}(\mathbf{r}) = \sum_{m,n} \mathbf{E}_{m,n} e^{-i(k_x x + k_y y)} e^{-i(m\beta_0 x + n\beta_0 y)}. \tag{1}$$

Here, $\mathbf{E}_{m,n}$ is an electric field vector of each Fourier component ($m$, $n$ are integers), $\beta_0 = 2\pi/a$ is the magnitude of the reciprocal lattice vector, and $\mathbf{k} = (k_x, k_y)$ is a wavevector representing a deviation from the Γ point. Resonance at the Γ point, which is used for typical PCSELs, is composed of four fundamental waves expressed with $(m, n) = (\pm 1, 0), (0, \pm 1)$, where the complex

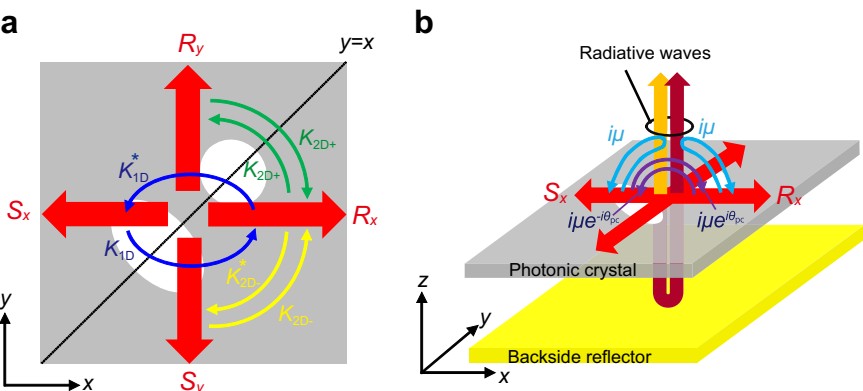

**Fig. 1 Hermitian and non-Hermitian optical couplings inside PCSELs. a** Hermitian couplings between the four fundamental waves ($R_x$, $S_x$, $R_y$, $S_y$) inside a PCSEL. **b** Non-Hermitian couplings via radiated waves, where a backside reflector is used for the control of the non-Hermitian coupling coefficient $i\mu$.

electric-field amplitudes of these waves are expressed as $R_x$, $S_x$, $R_y$, and $S_y$ as illustrated in Fig. 1a. These four waves are directly coupled with each other, and are also indirectly coupled via higher-order waves ($m^2 + n^2 > 1$) and radiative waves ($m = n = 0$). Considering the reflection symmetry along $y = x$ in the double-lattice photonic crystal as shown in Fig. 1, the mutual couplings between these fundamental waves can be expressed with the following matrix equations in the framework of the three-dimensional coupled-wave theory (3D-CWT)[17,18];

$$\left(\delta + i\frac{\alpha}{2}\right)\begin{pmatrix} R_x \\ S_x \\ R_y \\ S_y \end{pmatrix} = (\mathbf{C}_{\text{Hermitian}} + \mathbf{C}_{\text{non-Hermitian}} + \mathbf{C}_{\text{non-Gamma}})\begin{pmatrix} R_x \\ S_x \\ R_y \\ S_y \end{pmatrix},$$

(2)

$$\mathbf{C}_{\text{Hermitian}} = \begin{pmatrix} \kappa_{11} & \kappa_{1D} & \kappa_{2D+} & \kappa_{2D-} \\ \kappa_{1D}^* & \kappa_{11} & \kappa_{2D-}^* & \kappa_{2D+} \\ \kappa_{2D+} & \kappa_{2D-} & \kappa_{11} & \kappa_{1D} \\ \kappa_{2D-}^* & \kappa_{2D+} & \kappa_{1D}^* & \kappa_{11} \end{pmatrix},$$

(3)

$$\mathbf{C}_{\text{non-Hermitian}} = \begin{pmatrix} i\mu & i\mu e^{i\theta_{\text{pc}}} & 0 & 0 \\ i\mu e^{-i\theta_{\text{pc}}} & i\mu & 0 & 0 \\ 0 & 0 & i\mu & i\mu e^{i\theta_{\text{pc}}} \\ 0 & 0 & i\mu e^{-i\theta_{\text{pc}}} & i\mu \end{pmatrix},$$

(4)

$$\mathbf{C}_{\text{non-Gamma}} = \begin{pmatrix} \beta_{k;1,0} - \beta_0 & 0 & 0 & 0 \\ 0 & \beta_{k;-1,0} - \beta_0 & 0 & 0 \\ 0 & 0 & \beta_{k;0,1} - \beta_0 & 0 \\ 0 & 0 & 0 & \beta_{k;0,-1} - \beta_0 \end{pmatrix}$$

$$\sim \begin{pmatrix} k_x & 0 & 0 & 0 \\ 0 & -k_x & 0 & 0 \\ 0 & 0 & k_y & 0 \\ 0 & 0 & 0 & -k_y \end{pmatrix},$$

(5)

$$\beta_{k;m,n} = \sqrt{(k_x + m\beta_0)^2 + (k_y + n\beta_0)^2}.$$

Expressions for each term on the right-hand side of Eqs. (3) and (4) are provided in Supplementary Section 1. The real and imaginary part of the eigenfrequency ($\delta$ and $\alpha$) on the left side of Eq. (2) denote the wavenumber (frequency) and loss (radiation constant) of each resonant mode, respectively.

$\mathbf{C}_{\text{Hermitian}}$ in Eq. (3) is a Hermitian matrix, where the condition $\mathbf{C}_{\text{Hermitian}} = \mathbf{C}_{\text{Hermitian}}^\dagger$ is satisfied. $\mathbf{C}_{\text{Hermitian}}$ expresses the couplings among the fundamental waves (Fig. 1a), which consists of a 180° (or 1D) coupling coefficient ($\kappa_{1D}$), 90° (or 2D) coupling coefficients ($\kappa_{2D+}$, $\kappa_{2D-}$), and self-coupling coefficient ($\kappa_{11}$), where all the couplings do not accompany vertical emission loss. Note that $\kappa_{2D+}$ and $\kappa_{2D-}$ differ in value, because the double-lattice photonic crystal does not have $C_4$ symmetry unlike a general single-lattice photonic crystal with circular lattice points. We should also note that $\kappa_{1D}$ and $\kappa_{2D-}$ are complex numbers because the double-lattice structure does not have $C_2$ symmetry, while $\kappa_{2D+}$ is a real number because of the reflection symmetry about the line of $y = x$. $\kappa_{11}$ is a real number, which expresses self-coupling for fundamental four waves without accompanying vertical emission loss.

$\mathbf{C}_{\text{non-Hermitian}}$ in Eq. (4) shows non-Hermitian couplings of the fundamental waves through radiative waves, where the condition $\mathbf{C}_{\text{non-Hermitian}} = -\mathbf{C}_{\text{non-Hermitian}}^\dagger$ is satisfied. Note that in previous references on 3D-CWT[17,18], mutual couplings via radiative waves

were expressed with another coupled-wave matrix $\mathbf{C}_{\text{rad}}$. The difference between $\mathbf{C}_{\text{rad}}$ and $\mathbf{C}_{\text{non-Hermitian}}$ is that the former contains both non-Hermitian and Hermitian couplings, while the latter retains only non-Hermitian couplings. Such reconstruction of the coupled-wave matrices facilitates the derivation of analytical formulae of the radiation constants and threshold margin in a finite-sized PCSEL, as shown later. In $\mathbf{C}_{\text{non-Hermitian}}$, $i\mu$ is a purely imaginary number, which expresses self-coupling of four fundamental waves through radiative waves with accompanying vertical emission loss (Fig. 1b). The magnitude of $\mu$ can be continuously changed by changing the phase difference between the upward-radiated wave and the downward-radiated wave that is reflected at the bottom reflector. $i\mu e^{\pm i\theta_{\text{pc}}}$ expresses ±180° coupling through radiative waves with accompanying vertical emission loss. $\theta_{\text{pc}}$ represents the phase change associated with ±180° coupling [see Supplementary Eq. (S14) and Supplementary Fig. S1 in Supplementary Section 1 for details].

$\mathbf{C}_{\text{non-gamma}}$ in Eq. (5) denotes the deviation of the wavevector from the Γ point, which induces the change in frequency.

**Frequencies and radiation constants at the Γ point.** In this section, we consider a PCSEL with infinite size, for which the electric field distribution is periodic in the plane of a photonic crystal (the effect of a finite size is considered in the next section). For the infinite-size PCSEL, the loss originates from the vertical loss (non-Hermitian process) expressed by a radiation constant. Because there are four possible band-edge modes at the Γ point (A, B, C, D), we derive the radiation constants for these modes, together with their resonant frequencies by using Eq. (2).

Before doing so, we first consider only the Hermitian process expressed by the first term of the right hand side of Eq. (2) ($\mathbf{C}_{\text{Hermitian}}$), while ignoring the non-Hermitian process. In this case, the optical couplings for the four fundamental waves can be expressed as:

$$\left(\delta + i\frac{\alpha}{2}\right)\begin{pmatrix} R_x \\ S_x \\ R_y \\ S_y \end{pmatrix} = \begin{pmatrix} \kappa_{11} & \kappa_{1D} & \kappa_{2D+} & \kappa_{2D-} \\ \kappa_{1D}^* & \kappa_{11} & \kappa_{2D-}^* & \kappa_{2D+} \\ \kappa_{2D+} & \kappa_{2D-} & \kappa_{11} & \kappa_{1D} \\ \kappa_{2D-}^* & \kappa_{2D+} & \kappa_{1D}^* & \kappa_{11} \end{pmatrix}\begin{pmatrix} R_x \\ S_x \\ R_y \\ S_y \end{pmatrix}$$

(6)

This equation can be divided into the following equations by considering the reflection symmetry of the photonic crystal along $y = x$:

$$\left(\delta + i\frac{\alpha}{2}\right)\begin{pmatrix} R_x + R_y \\ S_x + S_y \end{pmatrix} = \begin{pmatrix} \kappa_{11} + \kappa_{2D+} & \kappa_{1D} + \kappa_{2D-} \\ \kappa_{1D}^* + \kappa_{2D-}^* & \kappa_{11} + \kappa_{2D+} \end{pmatrix}\begin{pmatrix} R_x + R_y \\ S_x + S_y \end{pmatrix},$$

(7)

$$\left(\delta + i\frac{\alpha}{2}\right)\begin{pmatrix} R_x - R_y \\ S_x - S_y \end{pmatrix} = \begin{pmatrix} \kappa_{11} - \kappa_{2D+} & \kappa_{1D} - \kappa_{2D-} \\ \kappa_{1D}^* - \kappa_{2D-}^* & \kappa_{11} - \kappa_{2D+} \end{pmatrix}\begin{pmatrix} R_x - R_y \\ S_x - S_y \end{pmatrix}.$$

(8)

The physical meanings of the above equations are shown in Fig. 2. Figure 2a shows the electric field vectors of the four fundamental waves in a general case, which are coupled with each other according to Eq. (6), where $\mathbf{e}_x$ and $\mathbf{e}_y$ are unit vectors in the $x$ and $y$ directions, respectively. Following the division of Eq. (6) into Eq.(7) and Eq. (8), Fig. 2a can be divided into Fig. 2b and Fig. 2c; Fig. 2b corresponds to Eq. (7), where the electric-field pairs that have anti-symmetric vectors about the axis of $y = x$ with the amplitudes of $R_x + R_y$ and $S_x + S_y$ are coupled with each other, while Fig. 2c corresponds to Eq. (8), where the electric-field pairs that have symmetric vectors about the axis of $y = x$ with the amplitudes of $R_x - R_y$ and $S_x - S_y$ are coupled with each other. The coupling coefficients between the anti-

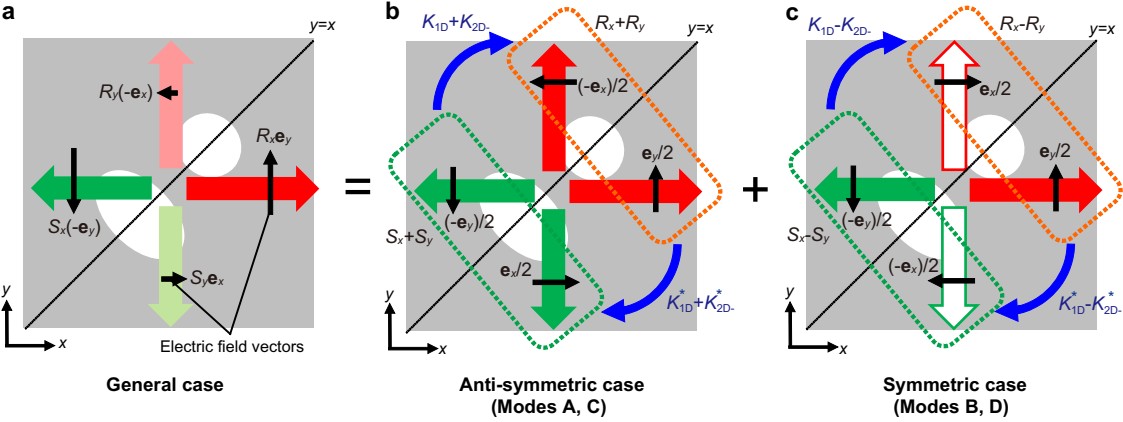

**Fig. 2 Physical meaning of Hermitian coupling coefficients. a** Electric field vectors of the four fundamental waves in a general case. **b** Electric field vector pairs that have anti-symmetric vectors about the axis of $y = x$ with the amplitudes of $R_x + R_y$ and $S_x + S_y$, where their Hermitian coupling coefficients are expressed as $\kappa_{1D} + \kappa_{2D-}$ and $\kappa_{1D}^* + \kappa_{2D-}^*$. The resultant anti-symmetric modes are defined as modes A and C. **c** Electric field vector pairs that have symmetric vectors about the axis of $y = x$ with the amplitudes of $R_x - R_y$ and $S_x - S_y$, where their Hermitian coupling coefficient are expressed as $\kappa_{1D} - \kappa_{2D-}$ and $\kappa_{1D}^* - \kappa_{2D-}^*$. The resultant symmetric modes are defined as modes B and D.

symmetric electric-field pairs ($R_x + R_y$ and $S_x + S_y$) in Fig. 2b are represented by $\kappa_{1D} + \kappa_{2D-}$ and $\kappa_{1D}^* + \kappa_{2D-}^*$ as shown in the non-diagonal terms in Eq. (7), and thus, $\kappa_{1D} + \kappa_{2D-}$ represents the Hermitian coupling coefficient for the anti-symmetric modes (which we define as modes A and C). Similarly, the coupling coefficients between the symmetric electric-field pairs ($R_x - R_y$ and $S_x - S_y$) in Fig. 2c are represented by $\kappa_{1D} - \kappa_{2D-}$ and $\kappa_{1D}^* - \kappa_{2D-}^*$, and thus $\kappa_{1D} - \kappa_{2D-}$ represents the Hermitian coupling coefficient for the symmetric modes (which we define as modes B and D).

Now, we introduce the non-Hermitian process in addition to the Hermitian process to obtain the radiation constants for the infinite photonic crystal structure. If we consider the non-Hermitian coupling coefficients ($i\mu$ and $i\mu e^{\pm i\theta_{pc}}$) in Eq. (4) together with the aforementioned Hermitian coupling coefficients in Eq. (3), the analytical formulae of the complex eigenfrequencies, which express the resonant frequencies $\delta_i$ and radiation constants $\alpha_i$, of the four modes at the $\Gamma$ point ($i$ = A, B, C, D) can be derived as follows (the detailed derivation of these formulae are provided in Supplementary Section 2).

For modes A and C:

$$\begin{aligned}
\delta_{A,C} + i\alpha_{A,C}/2 &= \kappa_{11} + \kappa_{2D+} + i\mu \\
&\mp \sqrt{[(\kappa_{1D} + \kappa_{2D-}) + i\mu e^{i\theta_{pc}}][(\kappa_{1D} + \kappa_{2D-})^* + i\mu e^{-i\theta_{pc}}]} \\
&= \kappa_{11} + \kappa_{2D+} + i\mu \\
&\mp \sqrt{[(\kappa_{1D} + \kappa_{2D-})e^{-i\theta_{pc}} + i\mu][\{(\kappa_{1D} + \kappa_{2D-})e^{-i\theta_{pc}}\}^* + i\mu]},
\end{aligned} \tag{9}$$

and, for modes B and D:

$$\begin{aligned}
\delta_{B,D} + i\alpha_{B,D}/2 &= \kappa_{11} - \kappa_{2D+} + i\mu \\
&\mp \sqrt{[(\kappa_{1D} - \kappa_{2D-}) + i\mu e^{i\theta_{pc}}][(\kappa_{1D} - \kappa_{2D-})^* + i\mu e^{-i\theta_{pc}}]} \\
&= \kappa_{11} - \kappa_{2D+} + i\mu \\
&\mp \sqrt{[(\kappa_{1D} - \kappa_{2D-})e^{-i\theta_{pc}} + i\mu][\{(\kappa_{1D} - \kappa_{2D-})e^{-i\theta_{pc}}\}^* + i\mu]}.
\end{aligned} \tag{10}$$

In Eqs. (9) and (10), the Hermitian coupling coefficients $\kappa_{1D} + \kappa_{2D-}$ and $\kappa_{1D} - \kappa_{2D-}$ in Eqs. (7) and (8) are rewritten as $(\kappa_{1D} + \kappa_{2D-})e^{-i\theta_{pc}}$ and $(\kappa_{1D} - \kappa_{2D-})e^{-i\theta_{pc}}$ respectively. These modifications allow us to consider the relative phase of $\kappa_{1D} + \kappa_{2D-}$ and $\kappa_{1D} - \kappa_{2D-}$ with respect to the phase of non-Hermitian $\pm180°$-coupling $\theta_{pc}$, wherein the coefficients are made invariant with respect to global translation of the air holes inside the unit cell (see Supplementary Fig. S1c in the Supplementary Section 1 for details). Hereafter, we call $(\kappa_{1D} + \kappa_{2D-})e^{-i\theta_{pc}}$ the "phase-invariant effective Hermitian coupling coefficient" for modes A and C, and

we call $(\kappa_{1D} - \kappa_{2D-})e^{-i\theta_{pc}}$ the phase-invariant effective Hermitian coupling coefficient for modes B and D.

Next, we consider the specific case of $|(\kappa_{1D} + \kappa_{2D-})e^{-i\theta_{pc}}| = |\kappa_{1D} + \kappa_{2D-}| \sim 0$, which corresponds to the case in which destructive interference of not only 180° diffraction but also 90° diffraction is achieved, as we discussed in our previous paper[14]. In this case, the radiation constants of the four modes can be derived from Eqs. (9) and (10) as follows:

$$\alpha_A \sim 0 \text{ (controllable)}$$
$$\alpha_C \sim 4\mu \gg 0$$
$$\alpha_B \sim 2\mu - 2\left|\text{Im}\sqrt{(2\kappa_{1D}e^{-i\theta_{pc}} + i\mu)(2\kappa_{1D}^* e^{i\theta_{pc}} + i\mu)}\right| > 0 \tag{11}$$
$$\alpha_D \sim 2\mu + 2\left|\text{Im}\sqrt{(2\kappa_{1D}e^{-i\theta_{pc}} + i\mu)(2\kappa_{1D}^* e^{i\theta_{pc}} + i\mu)}\right| \gg 0.$$

In Eq. (11), it is seen that the radiation constant $\alpha_A$ of mode A is much smaller than those of the other modes (B, C, D). Specifically, the difference in the radiation constant between mode A and the other three modes can be shown to be sufficiently large ($>20$ cm$^{-1}$) (see Supplementary Fig. S2 in Supplementary Section 3 for detail), which indicates that the lasing oscillation occurs stably in mode A. Therefore, we hereafter focus on mode A along with mode C as its counterpart, since these two modes have the same symmetry as described above.

Next, we show that the value of $\alpha_A$ is controllable by changing the value of the phase-invariant effective Hermitian coupling coefficient $(\kappa_{1D} + \kappa_{2D-})e^{-i\theta_{pc}}$ around zero, in addition to the value of the non-Hermitian coupling coefficient $i\mu$. Toward this purpose, we define here the real and imaginary parts of $(\kappa_{1D} + \kappa_{2D-})e^{-i\theta_{pc}}$ as $R$ and $I$, respectively, and we assume that $|I|$ is much smaller than $|R + i\mu|$. The physical meaning of this assumption and the role of $I$ in radiation process are explained in Supplementary Section 4. Under this assumption, Eq. (9) for modes A and C can be then transformed as

$$\begin{aligned}
\delta_{A,C} + i\alpha_{A,C}/2 &= \kappa_{11} + \kappa_{2D+} + i\mu \mp \sqrt{(R + i\mu)^2 + I^2} \\
&\sim \kappa_{11} + \kappa_{2D+} + i\mu \mp (R + i\mu)\left[1 + \frac{I^2}{2(R + i\mu)^2}\right]
\end{aligned} \tag{12}$$

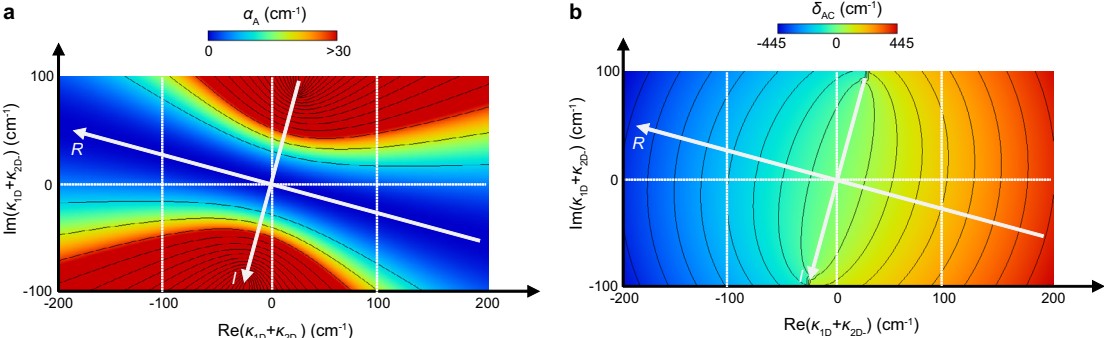

**Fig. 3 Control of radiation constant and frequency difference via Hermitian coupling coefficient. a** Radiation constant of mode A. The interval of the contour lines is 10 cm⁻¹. **b** Frequency gap $\delta_{AC}$ between modes A and C. The interval of the contour lines is 50 cm⁻¹.

Equation (12) gives the following analytical expressions for the frequency difference between modes A and C at the Γ point ($\delta_{AC}$) and the radiation constants ($\alpha_A$ and $\alpha_C$):

$$\delta_{AC} = \delta_A - \delta_C = -2R - \frac{R}{\mu^2 + R^2}I^2 \sim -2R \quad (13)$$

$$\alpha_A = \frac{\mu}{\mu^2 + R^2}I^2, \; \alpha_C = 4\mu - \alpha_A \quad (14)$$

It is seen in these equations that the frequency gap $\delta_{AC}$ between modes A and C is mostly determined by the real part of the phase-invariant effective Hermitian coupling coefficient, $R$, while $\alpha_A$ is determined by both real and imaginary parts of the phase-invariant effective Hermitian coupling coefficients, $R$ and $I$, as well as the absolute value of the non-Hermitian coupling coefficient $\mu$. Thus, $\alpha_A$ can be controlled by appropriately choosing the values of $R$, $I$, and $\mu$.

Figures 3a, b show examples of the radiation constant $\alpha_A$ and the mode gap frequency $\delta_{AC}$, respectively, which are calculated using Eq. (9), as functions of $\kappa_{1D} + \kappa_{2D-}$ in the complex plane, where $\mu$ is fixed to 90 cm⁻¹ ($\mu$ is also changed in the next section). Note that the axes of the phase-invariant effective Hermitian coupling coefficients $R$ and $I$ are also drawn in the figures; these axes are rotated with respect to those for $\kappa_{1D} + \kappa_{2D-}$ by $\theta_{pc}$, which is fixed to $0.92\pi$ as a typical value for the double-lattice structure whose midpoint is taken as the origin (see Supplementary Section 1). It is seen in Fig. 3a that the radiation constant $\alpha_A$ becomes 0 cm⁻¹ on the $R$ axis (namely, when $I = 0$ cm⁻¹), and this value can be set to a desirable value (from 0 cm⁻¹ to 30 cm⁻¹) by adjusting $|I|$. From this result, a photonic crystal with a double-lattice structure, or more generally a structure without $C_2$ symmetry, enables the flexible control of the radiation constant $\alpha_A$. It is also seen in Fig. 3b that the frequency gap $\delta_{AC}$ becomes 0 cm⁻¹ on the $I$ axis (namely, $R = 0$ cm⁻¹) in the range of $|I| \leq \mu$. The frequency degeneracy of the two bands A and C at the Γ point is due to the complete cancellation of the in-plane optical feedback.

On the other hand, unlike double-lattice photonic crystals, when a photonic crystal has a more general symmetry, such as $C_2$ or $C_4$ symmetry, $\kappa_{1D}$, $\kappa_{2D-}$ are purely real numbers and $\theta_{pc} = \pi$, so $I = 0$. In this case, Eqs. (9) and (10) can be transformed into the following simpler expressions:

$$\begin{aligned}
\delta_A &= \kappa_{11} + \kappa_{2D+} + \kappa_{1D} + \kappa_{2D-}, \; \alpha_A = 0 \\
\delta_C &= \kappa_{11} + \kappa_{2D+} - \kappa_{1D} - \kappa_{2D-}, \; \alpha_C = 4\mu \\
\delta_B &= \kappa_{11} - \kappa_{2D+} + \kappa_{1D} - \kappa_{2D-}, \; \alpha_B = 0 \\
\delta_D &= \kappa_{11} - \kappa_{2D+} - \kappa_{1D} + \kappa_{2D-}, \; \alpha_D = 4\mu.
\end{aligned} \quad (15)$$

The radiation constants of modes A and B are exactly zero, which prohibits laser emission in the vertical direction. Even when a small structural perturbation is added to the photonic crystal to enable vertical emission, the radiation constant difference between modes A and B remains small, which results in multiple-mode lasing.

**Threshold margin for single-mode lasing in a finite-sized PCSEL.** Next, we consider a finite-sized PCSEL and derive the conditions for increasing the threshold margin between the fundamental mode and higher-order modes originating from the same band-edge mode (mode A). As we explained in the previous section, the difference of radiation constant between mode A and the other three modes can be shown to be large enough (>20 cm⁻¹) to ensure that lasing occurs on band-edge A, and that the threshold margin between the fundamental mode and the higher-order modes within band-edge A is the "global threshold margin" of the large-area device (see Supplementary Section 5 for details). Figure 4a shows typical electric field distributions of the fundamental mode and the first higher-order mode inside a finite-sized device with a diameter of $L$, wherein the in-plane wavenumbers of the envelope functions of the two modes are approximately $\pi/L$ and $2\pi/L$, respectively[18,19] (the electric field distributions of other higher-order modes are shown in Supplementary Section 5). Therefore, by increasing the sensitivity of change of the radiation constant of mode A with respect to the in-plane wavenumbers (or $d\alpha_A/d\Delta k$), we can increase the threshold margin between the fundamental and higher-order modes. In the following analysis, we consider the band structure and radiation constants in the vicinity of the Γ point and derive the general conditions to maximize the threshold margin for single-mode lasing.

We first consider that the in-plane wavevector of the Bloch waves are slightly shifted along the $y = x$ axis of reflection symmetry (i.e., in the Γ-M direction, for which $k_x = k_y = \Delta k/\sqrt{2}$). By solving Eq. (2), we obtain the analytical formula of the complex eigenfrequencies as follows (see Supplementary Section 2);

$$\begin{aligned}
\delta_{A,C} + i\alpha_{A,C}/2 &= \kappa_{11} + \kappa_{2D+} + i\mu \\
&\mp \sqrt{(\kappa_{1D} + \kappa_{2D-} + i\mu e^{i\theta_{pc}})(\kappa_{1D}^* + \kappa_{2D-}^* + i\mu e^{-i\theta_{pc}}) + (\Delta k/\sqrt{2})^2} \\
&= \kappa_{11} + \kappa_{2D+} + i\mu \mp \sqrt{(R + i\mu)^2 + I^2 + (\Delta k/\sqrt{2})^2},
\end{aligned} \quad (16)$$

where not only mode A but also mode C are considered as the counterpart. Figures 4b and 4c show the calculated frequencies ($\delta_A$ and $\delta_C$) and radiation constants ($\alpha_A$ and $\alpha_C$) as functions of $\Delta k$, where $R$ is taken as a parameter and $\mu$ and $I$ are fixed to the

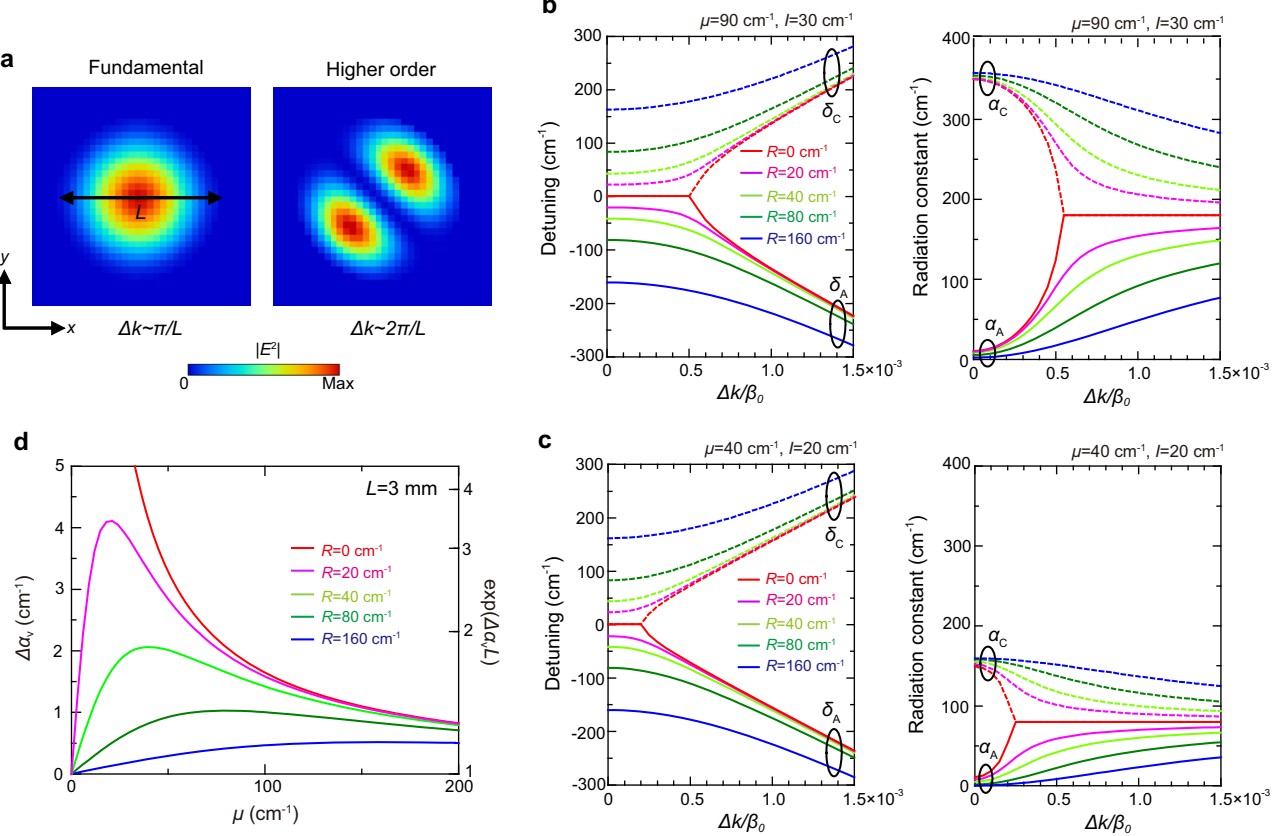

**Fig. 4 Radiation constant difference in finite-size PCSELs. a** Typical electric field distributions of the fundamental and higher-order modes in a finite-sized device with a diameter of $L$. **b, c** Calculated frequencies and radiation constants of modes A and C as functions of $\Delta k$ for several values of $R$, where the other parameters ($\mu$ and $I$) are fixed to constant values ($\mu = 90$ cm$^{-1}$, $I = 30$ cm$^{-1}$ in Fig. 4b and $\mu = 40$ cm$^{-1}$, $I = 20$ cm$^{-1}$ in Fig. 4c). **d** Threshold margin ($\Delta\alpha_{\rm v}$ and $e^{\Delta\alpha_{\rm v}\cdot L}$) for a 3-mm-diameter PCSEL when the values of $R$ and $\mu$ are changed.

constant values ($\mu = 90$ cm$^{-1}$, $I = 30$ cm$^{-1}$ in Fig. 4b and $\mu = 40$ cm$^{-1}$, $I = 20$ cm$^{-1}$ in Fig. 4c). As seen in the left panels of Figs. 4b and 4c, the frequency gap $\delta_{\rm AC}$ between modes A and C decreases when $R$ decreases [see also Eq. (13)], while the change in the radiation constant $\alpha_{\rm A}$ (and $\alpha_{\rm C}$) becomes steeper when $R$ decreases (see the right panels). Moreover, the change in $\alpha_{\rm A}$ and $\delta_{\rm AC}$ becomes more sensitive to a change of $\Delta k$ in the vicinity of the $\Gamma$ point especially for the smaller non-Hermitian coupling coefficient $\mu$ (Fig. 4c). Although the most sensitive change can be obtained at $R = 0$ cm$^{-1}$, where both the frequencies and the radiation constants of modes A and C degenerate at $\Delta k = \pm\sqrt{2(\mu^2 - I^2)}$, forming an exceptional point[20,21], this point should be avoided for stable single-mode oscillation as shown later.

The radiation constant $\alpha_{\rm A}$ of mode A in the vicinity of the $\Gamma$ point and under the condition of ($\sqrt{I^2 + (\Delta k/\sqrt{2})^2} < \sqrt{R^2 + \mu^2}$), Eq. (16) can be approximated as follows;

$$\alpha_{\rm A,\Delta k} \sim \frac{\mu}{\mu^2 + R^2}[I^2 + (\Delta k/\sqrt{2})^2]. \tag{17}$$

The first term of the right side of Eq. (17) denotes the radiation constant equal to Eq. (14) that corresponds to that of the infinite size PCSEL, while the second term denotes the increase of the radiation constant owing to the deviation from the $\Gamma$ point due to the finite-size effect. Using Eq. (17), the radiation constant difference between the fundamental mode and the 1st higher-order mode (shown in Fig. 4a) can be expressed with the

following simple formula:

$$\Delta\alpha_{\rm v} = \alpha_{\rm A,\Delta k=2\pi/L} - \alpha_{\rm A,\Delta k=\pi/L} \sim \frac{\mu}{\mu^2 + R^2}\frac{3\pi^2}{2L^2}. \tag{18}$$

Although Eq. (18) is derived for the higher-order mode in the $\Gamma$-M direction, it can be approximately applied to the higher-order mode in its orthogonal direction ($\Gamma$-M') because the band structures in the $\Gamma$-M direction and $\Gamma$-M' direction are almost equal in the vicinity of the $\Gamma$ point (see Supplementary Section 2). It should be also noted that the threshold margin between the fundamental mode and the higher-order modes depends not only on $\Delta\alpha_{\rm v}$ in Eq. (18) but also on the difference of their in-plane losses ($\Delta\alpha_{//}$). However, the contribution of the former is dominant in the case of large-area PCSELs ($L \geq 3$ mm) because the portion of the electric field penetrating outside the active region becomes small (see Supplementary Section 6). We should note that the threshold margin might be better expressed by the exponential of the product of $\Delta\alpha_{\rm v}$ in Eq. (18) and the device diameter $L$:

$$e^{\Delta\alpha_{\rm v}\cdot L} \sim \exp\left(\frac{\mu}{\mu^2 + R^2}\frac{3\pi^2}{2L}\right) \tag{19}$$

This is because $e^{\Delta\alpha_{\rm v}\cdot L}$ directly express the ratio between the light amplification rate of the fundamental mode and the 1st higher-order mode when the light propagates from one edge to the other edge of the device.

The calculated threshold margins [$\Delta\alpha_{\rm v}$ in Eq. (18) and $e^{\Delta\alpha_{\rm v}\cdot L}$ in Eq. (19)] are shown in Fig. 4d for PCSELs with $L = 3$ mm for

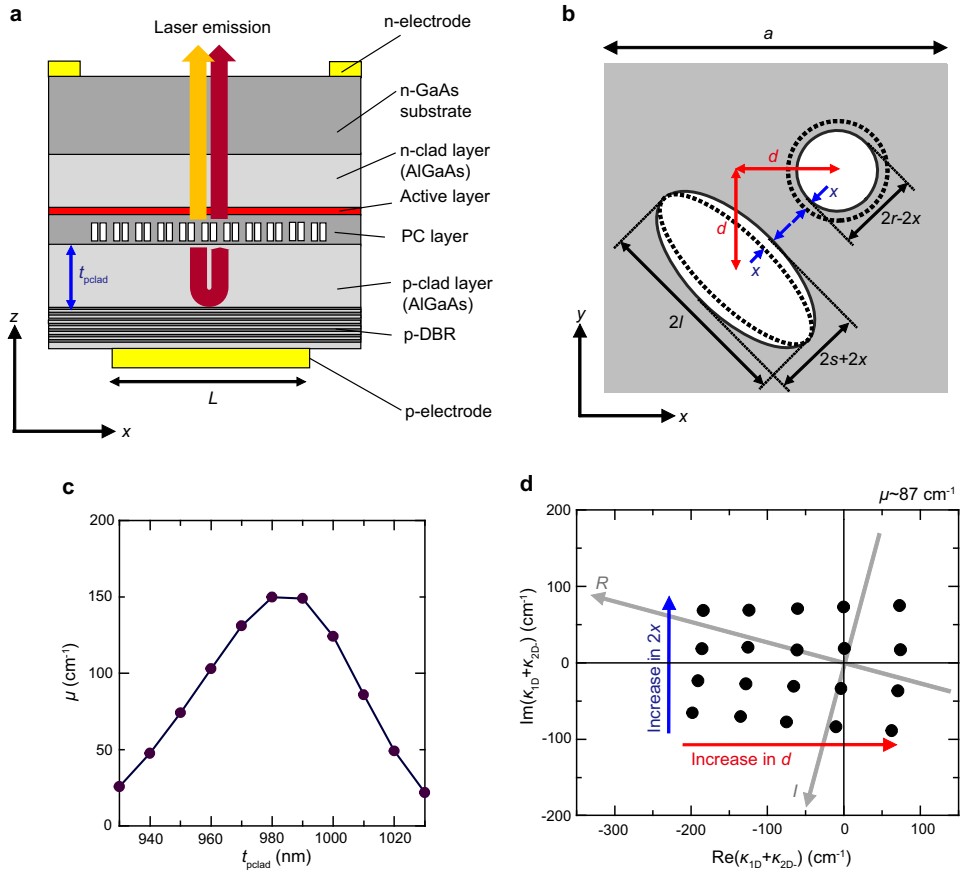

**Fig. 5 Double-lattice PCSEL with Distributed Bragg Reflector (DBR) for arbitrary control of coupling coefficients. a** Cross section of a double-lattice PCSEL with a DBR. **b** Top view of the double-lattice photonic crystal composed of an elliptical and circular hole. $d$ is the lattice separation and $2x$ is the tuning parameter for the hole sizes (or balance of large and small holes). The details of the parameters are explained in Supplementary Section 8.
**c** Magnitude of non-Hermitian coupling coefficient $\mu$ as a function of the p-clad thickness $t_{pclad}$. **d** Calculated Hermitian coupling coefficient ($\kappa_{1D} + \kappa_{2D-}$ or $R$ and $I$) in the complex plane, when $d$ and $2x$ are varied.

various $R$ and $\mu$. It is seen that $\Delta\alpha_v$ (and $e^{\Delta\alpha_v \cdot L}$) increase as $R$ decreases, which can be ascribed to the destructive interference of not only 180° but also 90° diffractions and weakened in-plane optical feedback inside the photonic crystal. In addition, $\Delta\alpha_v$ (and $e^{\Delta\alpha_v \cdot L}$) are maximized when the absolute value of non-Hermitian coupling coefficient ($\mu$) balances with the real part of the phase-invariant effective Hermitian coupling coefficient ($R$) for a constant $R$. It should be noted that the value of $\mu$ should have a certain value to ensure a sufficient radiation constant difference among the four band-edge modes (A~D) in Eq. (11). Therefore, for the realization of single-mode lasing operation in an ultra-large-area PCSEL while keeping the large threshold margin, it is important to manipulate not only the phase-invariant effective Hermitian optical couplings ($R$ and $I$) but also the non-Hermitian optical couplings ($i\mu$) by controlling both the lattice-point design and the complex reflectivity of the backside reflector (shown in Fig. 1).

Although Eqs. (18) and (19) provide a general guideline for increasing the threshold margin between the fundamental mode and the 1st higher-order mode inside PCSELs, its derivation does not consider the carrier-photon interactions and the spatial non-uniformity of the carrier distribution inside the device, which is inevitably caused by the spatial hole burning effect at high current injection levels. Therefore, in the next section, we design a concrete photonic crystal structure appropriate for ultra-large-area ($L = 3{-}10$ mm) PCSELs with

various Hermitian and non-Hermitian optical coefficients, and we discuss the lasing stability of the designed PCSELs through a comprehensive analysis of lasing characteristics considering the carrier-photon interactions.

**Concrete design of single-mode ultra-large-area PCSELs**. Figures 5a, b show the cross section and the top view of the designed PCSEL for single-mode ultra-large-area operation. In Fig. 5a, a GaAs photonic crystal layer is placed near the active layer (InGaAs/AlGaAs quantum wells) and is sandwiched by p-clad and n-clad layers. The light emitted downward from the photonic crystal layer is reflected at the p-type distributed Bragg reflector (DBR) below the p-clad layer and then interferes with the upward emission. Figure 5b shows a double-lattice structure composed of an elliptic and circular hole, where the tuning parameters are a lattice separation of $d$ and hole-size balance $2x$.

The magnitude of the non-Hermitian coupling coefficient $\mu$ can be continuously changed by adjusting the phase difference between the upward and downward emission with the thickness of the p-clad layer (Fig. 5a). On the other hand, the real and imaginary parts of the Hermitian coupling coefficient $\kappa_{1D} + \kappa_{2D-}$ can be manipulated by tuning a lattice separation of $d$ and hole-size balance $2x$ in the double-lattice structure (Fig. 5b). The most dominant Fourier component that determines $\kappa_{1D} + \kappa_{2D-}$ is $\xi_{2,0}$ (see Supplementary Section 7), which induces the direct coupling

between $R_x$ and $S_x$ ($R_y$ and $S_y$). $\xi_{2,0}$ of a double-lattice photonic crystal with $d = 0.25a + \Delta d$ can be approximated as follows (see also Supplementary Section 7);

$$\xi_{2,0} \sim (n_{\text{GaAs}}^2 - n_{\text{air}}^2)\left[(FF_{\text{total}})\frac{2\pi}{a}\Delta d + i(\Delta FF)\right], \qquad (20)$$

where $n_{\text{GaAs}}^2 - n_{\text{air}}^2$ is the permittivity difference between GaAs and air, and $FF_{\text{total}}$ and $\Delta FF$ denote the sum and the difference of the filling factors of the two holes, respectively. As is apparent from Eq. (20), the double-lattice photonic crystal enables the independent control of the real and imaginary parts of $\xi_{2,0}$, and thus $\kappa_{1D} + \kappa_{2D-}$, by changing $\Delta d$ and $\Delta FF$ separately. More concretely, by changing $d$ and $2x$ in Fig. 5b, which induces the change in $\Delta d$ and $\Delta FF$, $\kappa_{1D} + \kappa_{2D-}$ (or $R$ and $I$) can be controlled in the entire complex plane.

We numerically calculate the magnitude of non-Hermitian coupling coefficient $\mu$ and the Hermitian coupling coefficient $\kappa_{1D} + \kappa_{2D-}$ (or $R$ and $I$) by changing the thickness of the p-clad layer $t_{\text{pclad}}$ and the structural parameters of the double-lattice photonic crystal ($d$, $2x$), respectively. The results are shown in Fig. 5c, d. The details of the simulation model and the parameters are given in Supplementary Section 8. As shown in these numerical simulations, the double-lattice photonic crystal resonators with backside reflectors allow the arbitrary control of the Hermitian and non-Hermitian coupling coefficients. It is worth emphasizing that the arbitrary control of $I$, which is enabled by the double-lattice structure, leads to the on-demand control of the radiation constant as shown in Fig. 3a.

**Lasing stability analysis considering carrier-photon interactions.** Finally, we analyze the lasing characteristics of the designed ultra-large-area PCSELs with the time-dependent 3D-CWT[22], which considers not only the mutual coupling of light but also the carrier-photon interactions and the spatial non-uniformity of the gain and refractive index distributions. The details of the simulation method are explained in Supplementary Section 8. We first consider 3-mm-diameter double-lattice PCSELs with a fixed p-clad thickness (which gives $\mu = 87$ cm$^{-1}$) and fixed hole sizes (which gives Im($\kappa_{1D} + \kappa_{2D-}$) $\sim -25$ cm$^{-1}$), and calculate the output power and lasing spectra by varying a lattice separation $d$ or the real part of the phase-invariant effective Hermitian coupling coefficient $R$. The above parameters are chosen so that a moderate radiation constant ($\sim 20$ cm$^{-1}$) is obtained for mode A while much higher radiation constants ($>40$ cm$^{-1}$) are obtained for the other band-edge modes.

Figure 6a, b show the calculated current-light-output (I-L) characteristics and lasing spectra of the four devices with different $R$, where the far-field beam pattern at each current is shown in the inset. In Fig. 6a, the threshold current and slope efficiency are almost the same for the designed devices, except for the one with a near-zero frequency gap between modes A and C ($R = -5$ cm$^{-1}$), which exhibits unstable lasing as explained later. The lasing spectra and the far-field beam patterns shown in Fig. 6b are completely different in the four devices. When $R = 86$ cm$^{-1}$, which almost equals $\mu$, single-mode lasing with a nearly diffraction-limited divergence angle ($\theta_{1/e}^2 \sim 0.03°$) is obtained at an injection current of 140 A, demonstrating the possibility of 100-W-class single-mode lasing in a PCSEL with a diameter as large as 3 mm. When $R = 178$ cm$^{-1}$, which is much larger than $\mu$, the lasing spectra broaden and the beam divergence angles increase, clearly showing the evidence of multimode lasing. This result agrees with the theoretical result shown in Fig. 4d, where the

threshold margin between the fundamental mode and the higher-order mode decreases as $R$ increases.

Interestingly enough, the broadening of the lasing spectra with the increase in the beam divergence also arises when $R = 25$ cm$^{-1}$, which is smaller than one third of $\mu$, and the spectra broaden even further for the device with a near-zero bandgap ($R = -5$ cm$^{-1}$). Such unstable lasing is caused by the carrier-induced refractive index change and the resultant frequency change inside the device. According to Eq. (16), the carrier-induced frequency change of mode A ($\Delta\delta_A/\Delta N$) and the carrier-induced radiation constant change $\Delta\alpha_A/\Delta N$ are related as follows (see Supplementary Section 9);

$$\frac{\Delta\alpha_A/\Delta N}{\Delta\delta_A/\Delta N} = -\frac{2\mu}{R}. \qquad (21)$$

From this equation, one can understand that when $R$ becomes much smaller than $\mu$, the change in the radiation constant $\Delta\alpha_A/\Delta N$ becomes more drastic than the change in the frequency $\Delta\delta_A/\Delta N$. For example, let us consider the case where the photon density locally decreases from the steady-state value. In a normal case, the photon density returns to the steady-state value immediately. This is because the local decrease of photon density induces the local increase of carrier density, which leads to the increase of local optical gain that increases the photon density again. However, for a device with a smaller $R$, the carrier-induced local frequency change causes the drastic increase in the radiation loss through Eq. (21), thereby leading to the further reduction of the local photon density and to the unstable oscillation. Therefore, for realizing stable single-mode lasing in the entire device, it is important not only to increase the threshold margin according to Eq. (18) [or Eq. (19)] but also to balance the real part of the phase-invariant effective Hermitian coupling coefficient and the magnitude of non-Hermitian coupling coefficient ($R\sim\mu$). This fact indicates that the ultimate case of $R = 0$ cm$^{-1}$, which forms the exceptional point discussed before, is not appropriate for stable lasing oscillation.

Based on the above discussion of the lasing stability, we finally investigate the feasibility of higher-power single-mode lasing in an even larger-size PCSEL ($L = 10$ mm). Here, we consider two different designs: (1) $R = 86$ cm$^{-1}$, $\mu = 87$ cm$^{-1}$, $I = 54$ cm$^{-1}$ and (2) $R = 45$ cm$^{-1}$, $\mu = 44$ cm$^{-1}$, $I = 32$ cm$^{-1}$. The former structure is the same as the one which enables single-mode lasing in a 3-mm-diameter PCSEL in Fig. 6b, and the latter structure has a larger threshold margin owing to the smaller $R$ and $\mu$. The calculated lasing spectra and the far-field beam patterns for the two structures are shown in Fig. 6c. While multimode lasing occurs in the former structure, single-mode lasing with a divergence angle $\theta_{1/e}^2 < 0.01°$ is obtained in the latter structure. The calculated I-L characteristic for the latter device is shown in Fig. 6d, demonstrating the feasibility of kW-class single-mode lasing in a centimeter-size PCSEL. The potential challenges and solutions for experimentally realizing the 100-W-to-1-kW PCSELs are discussed in Supplementary Section 10.

## Discussion

We have analytically provided general formulae for the complex eigenfrequencies of the four photonic bands around the $\Gamma$ point and derived the general conditions for ultra-large-area single-mode operation in PCSELs. We have proven that the threshold margin between the fundamental mode and the higher-order modes can be increased through the reasonable reduction of both the real part of phase-invariant effective Hermitian coupling coefficient and the magnitude of non-Hermitian coupling coefficients ($R$ and $\mu$), while

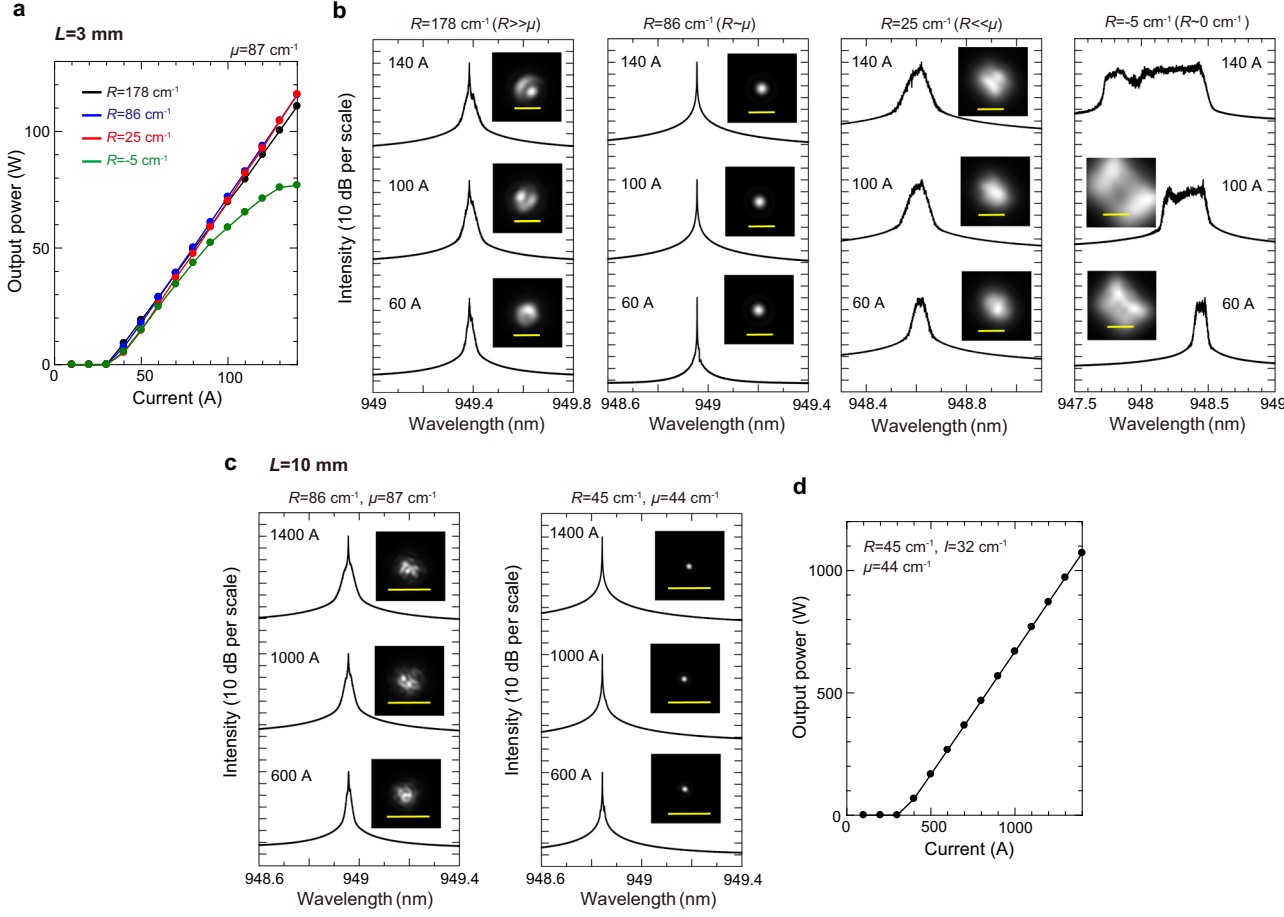

**Fig. 6 Numerical demonstration of single-mode lasing in ultra-large-area PCSELs with controlled coupling coefficients. a** Calculated I-L characteristics of 3-mm-diameter PCSELs with four different lattice separations $d$ or the real part of phase-invariant effective Hermitian coupling coefficient $R$. The $p$-clad thickness and hole sizes are fixed [$\mu = 87$ cm$^{-1}$]. The values of $I$ for the four devices are 72 cm$^{-1}$ (black), 54 cm$^{-1}$ (blue), 34 cm$^{-1}$ (red), and 34 cm$^{-1}$ (green), respectively. **b** Calculated lasing spectra and far-field beam patterns for the four devices. The yellow bar in each figure indicates a divergence angle of 0.05°. When $R$ almost equals $\mu$, single-mode lasing with a nearly diffraction-limited divergence angle is obtained. **c** Calculated lasing spectra and far-field beam patterns for 10-mm-diameter PCSELs with two different designs: $R = 86$ cm$^{-1}$, $\mu = 87$ cm$^{-1}$, $I = 54$ cm$^{-1}$ and $R = 45$ cm$^{-1}$, $\mu = 44$ cm$^{-1}$, $I = 32$ cm$^{-1}$. The yellow bar in each figure indicates a divergence angle of 0.05°. **d** Calculated I-L characteristic of the latter device ($R = 45$ cm$^{-1}$, $\mu = 44$ cm$^{-1}$, $I = 32$ cm$^{-1}$), which demonstrates the possibility of kW-class single-mode lasing in a centimeter-size PCSEL.

the balance between the two coefficients should be maintained to ensure stable lasing. In this context, it is shown that in the case of $R = 0$ cm$^{-1}$, where an exceptional point appears, the device performance becomes unstable owing to the carrier-induced refractive index change. Through the detailed numerical simulations, we have demonstrated that PCSELs with double-lattice photonic crystals and backside reflectors allow the arbitrary control of both Hermitian and non-Hermitian optical coupling coefficients, enabling 100-W-to-1-kW-class single-mode lasing with an ultra-large lasing diameter (≥3~10 mm). Our results provide universal guidelines towards the realization of one-chip kW-class next-generation semiconductor lasers, which are expected to replace conventional bulky high-power lasers, such as gas lasers, solid-state lasers and fiber lasers. Such ultra-compact high-power semiconductor lasers will bring innovation to a wide variety of industries using lasers, such as material processing[1,2], mobility[4], medicine[23], and even aerospace[24]. Our theoretical analysis, which considers not only Hermitian optical couplings but also non-Hermitian ones, also enables the detailed analysis of photonic bands around frequency gaps and exceptional points, which are attracting increasing

attention in non-Hermitian photonics[25,26]. We believe that the theory established in this work will contribute to the development of a wide variety of research fields from fundamental laser physics and non-Hermitian wave physics in general to industrial applications.

## Data availability

The data that supports the plots within this paper and other findings of this study are available within this article and its Supplementary Information files, and are also available from the corresponding author upon reasonable request.

## Code availability

The mathematical formulae of 3D-CWT simulations are available within this article and its Supplementary Information files, and their associated codes are available from the corresponding author upon reasonable request.

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

## Acknowledgements

This work was carried out under the project of Council for Science, Technology and Innovation (CSTI), Cross ministerial Strategic Innovation Promotion Program (SIP), "Photonics and Quantum Technology for Society 5.0" (Funding agency: QST) (S.N.). This work was partially supported by a Grant-in-Aid for Scientific Research [22H04915 (S.N.), 20H02655 (T.I.)] from the Japan Society for the Promotion of Science (JSPS).

## Author contributions

S.N. supervised the entire project. T.I. established the theory with M.Y., J.G., and K.Y.; T.I. performed the numerical simulations with M.Y., J.G., and K.I.; S.N. and T.I. discussed the results and wrote the paper with J.G., M.Y., K.I., and M.D.Z.

## Competing interests

The authors declare no competing interests.
