## [Peer review file · Nature Communications]

REVIEWER COMMENTS

Reviewer #1 (Remarks to the Author):

The authors determine the general conditions for high power single-mode operation in Photonic Crystal Surface Emitting Lasers. They consider both Hermitian and non-Hermitian optical coupling inside PCSELS, and derive radiation differences between the fundamental and higher-order modes, and describe suitable photonic crystal structures.

The work is of high quality and the key aspects of the PCSEL are considered.

The manuscript is somewhat unusual for Nature Communications. This work will be important and contains valuable advances for the PCSEL community. But while PCSELS are important devices, the interest in this paper will be limited outside of this field.

Reviewer #2 (Remarks to the Author):

In this manuscript, the authors systematically analyzed the performance of double-lattice PCSEL designs, mainly on the aspect of single-mode stability. Although the overall paper is based on theoretical computation without experimental validation, the analysis is thorough and solid. They explicitly pointed out a design guideline towards the next generation of PCSELS with optimal single-mode emission power potentially up to over 100 W. The mathematical tools used by the authors, i.e. the 3D time-dependent coupled wave theory model, exhibits its superior functionality in guiding PCSEL designs, proved by its capability of predicting the IV curves, spectra and far-field patterns at varying injection currents. Furthermore, the authors' analysis on the lasing stability of the accidentally degenerate exceptional point is valuable. From the aspect of spatial hole burning, they point out that such exceptional points are not ideal as lasing candidates due to the unstable oscillation. Although the authors didn't explicitly emphasize, this claim is different from some previous literature, such as [Opt. Lett. 39, 2072 (2014)], and will be helpful and inspiring for other groups in the community of photonic crystals and PCSELS.

However, there are some aspects in this manuscript where I think could be better optimized. I suggest the manuscript be accepted after addressing the following comments:

1. Generally, the presentation of the Coupled wave theory is quite mathematical in this paper. In addition to the variables defined in previous publications of the authors (R_x , S_x , R_y , S_y , κ_{1D} , κ_{2D}), they sequentially introduced several new mathematical terms in this paper, such as: μ (line 127), θ_{pc} (line 132), $\kappa_{1D} + \kappa_{2D}$ (line 163), R , I (lines 179, 180). However,

the physical meanings of these parameters are, to me, rather abstract and obscure. For example, $\kappa_{1D} + \kappa_{2D}$ is defined as the “effective Hermitian coupling coefficient” (line 163). In some sections, the authors analyzed the model by tuning the real and the imaginary parts of this coefficient (Fig. 2). In some other sections, the analysis is then based on the “phase-invariant effective Hermitian coupling coefficient” which is essentially the previous coefficient multiplied by a phase term $\exp{-i\theta_{pc}}$ (Fig. 3). Indeed, the model itself is complete and strict. However, a direct and clear expression of the physical nature of the CWT, especially in explaining the single-mode stability, might be hindered by such a way of presentation. And the readers might find it difficult to follow the authors’ framework, or the physical intuition. If possible, I suggest the authors re-express the model in a more compact way and explain the defined parameters with clear and explicit physical meanings. I believe by doing so, the potential impact of this manuscript could be enhanced. Specifically, it is not clear to me why the term $\kappa_{1D} + \kappa_{2D}$ (line 163) is a specially “effective” coefficient. The definition of θ_{pc} (line 132) is given in the Supplementary, yet the physical meaning in terms of “phase” is not clearly explained. Consequently, I don’t fully understand why $(\kappa_{1D} + \kappa_{2D})\exp{-i\theta_{pc}}$ is then “phase-invariant”. I am confused about the necessity of defining both the “effective coefficient” and the “phase-invariant” version of this coefficient at the same time. The difference in the physical meanings of them is not clearly explained. Furthermore, I understand that the term μ can be tuned by adjusting the cladding thickness, but I’m not clear how θ_{pc} can be independently tuned by adjusting the geometrical parameters of the PCSEL.

2. After reading the manuscript, I understood one novelty in this manuscript is the consideration of the “non-Hermitian optical couplings inside PCSELs” (line 62, lines 126-134). However, I later find the non-Hermitian coupling matrix $C_{\text{non-Hermitian}}$ (Eq. 4) is very similar to the C_{rad} defined in previous publication (eg. A13 of [PRB 99, 035308 (2019)], A17 of [PRB 84, 195119 (2011)]). To avoid the potential misunderstanding of readers like me, it might be helpful for the authors to clarify the difference in the exact physics and implementation of CWT in this manuscript to the previously published ones, except for the newly defined parameters as mentioned in comment 1.

3. At lines 172 and Fig. 2, the authors choose mode C as the counterpart of target mode A. However, it is not clear from Eq. 8 that the loss of mode B is necessarily larger than that of mode C. In addition, the authors also mentioned that in the highly symmetric cases, the radiation losses of mode B is zero (Eq. 12, line 217). Therefore, I wonder what is the result if also taking mode B as the counterpart of mode A in a similar analysis.

4. At line 180, the authors assume that “ $|I|$ is much smaller than $|R+i\mu|$ ”. Could they explain the physical meaning of such an assumption?

5. At line 205, the authors mention that modes A and C are degenerate at “ Γ point”. I wonder what is the definition of the “ Γ point” here - is it the point of $R=I=0$ in Fig. 2, or the generally accepted definition of $k=0$ point in a band structure? If the latter, I think it helps to show the band structure somewhere (maybe in the Supplementary). If the former, please clarify it.

6. I’m confused at the figure of merit in optimising the single-mode stability. In Fig. 3b and lines 237-249, the authors are emphasising on the “sensitivity of change in α_A and δ_{AC} with respect to Δk ”. However, why not the figure of merit as “the maximum difference in the loss for all Δk ”? After all, this is closer to the difference in lasing threshold between the modes, which is related to the

single-mode stability in a more direct way. As shown in the right panel of Fig. 3b, the case of $R=160\text{cm}^{-1}$ allows the maximum difference between α_A and α_C for all Δ_k , yet this is the case which is the least steep.

7. In pages 13-15, the authors considered the cases with Δ_K only in Γ -M and Γ -M' directions. What about Γ -X directions, or other arbitrary directions in the vicinity of the Γ point? The choice of selection here should be better justified that it would not lose the generality.

8. The “threshold margin” - Δ_{α_v} (Eq. 15, Fig. 3c) only considered the radiation loss difference between the fundamental mode and its own high order mode, without considering other fundamental modes. To me, this is not fully precise and strict. Firstly, although for square-lattice PC, there are only 4 modes in the infinite case near Γ_2 point, however there are 6 fundamental modes in the finite case, as mentioned by an early publication of the authors' group [Sec. 3.2, Opt. Express 15, 3981-3990 (2007)], where fundamental modes O and O' are seemingly exclusive to finite structures. Secondly, as shown in Fig. 2i of a prior publication from the authors' group [Opt. Express 20, 15945(2012)], although the lowest loss mode there is the fundamental mode A, the one with the second lowest loss is not its first high-order mode, but modes from other groups. This is also the case in other publications, such as Fig. 2b of [Appl. Phys. Lett. 104, 021102 (2014)], Fig. 3a of [Opt. Express 15, 3981-3990 (2007)]. Therefore, I think it is more convincing to show a mode spectrum (α_L vs Δ_L) as these mentioned figures in prior publications, where the overall cavity losses of all finite fundamental modes and their high order modes can be clearly seen. And the authors could define another “global threshold margin” as the difference in threshold between the lowest two modes in such a figure, then analyse its robustness. Such a “global” analysis would be highly convincing and helpful in support to the conclusions of this manuscript.

9. As essentially a proposal, the authors could also discuss the potential challenges in experimentally realizing the 100W or 1kW PCSELS proposed in this manuscript, if there is any. For example, would the current injection, or the thermal dissipation be a problem? If yes, how can they be overcome? Are there any other technical difficulties?

10. I think it helps in Fig. 5b to label also the relative relationship between R and μ in each panel. For example, “ $R \sim \mu$ ” could be labeled to the second panel.

11. I feel that there might be a typo in Fig. 1a - is a star sign (*) missing for one of the two “ κ_{2D+} ” terms?

12. Maybe the authors could briefly mention the potential impact of this manuscript to other groups outside the PCSEL community somewhere in the manuscript or in the Supplementary.

Response to the reviewers' comments (ID: NCOMMS-21-36910-T)

We are grateful to the two reviewers for their positive evaluation of our work and their useful suggestions that have helped us to improve our paper. As indicated in the response that follows, we have addressed all the concerns and suggestions of the reviewers in the revised version of our paper (Manuscript_revised_inoue.docx). The newly included sentences are shown in blue in the revised manuscript.

Reply to Reviewer #1

General Comment

The authors determine the general conditions for high power single-mode operation in Photonic Crystal Surface Emitting Lasers. They consider both Hermitian and non-Hermitian optical coupling inside PCSELS, and derive radiation differences between the fundamental and higher-order modes, and describe suitable photonic crystal structures.

The work is of high quality and the key aspects of the PCSEL are considered.

The manuscript is somewhat unusual for Nature Communications. This work will be important and contains valuable advances for the PCSEL community. But while PCSELS are important devices, the interest in this paper will be limited outside of this field.

Reply

We are grateful to the reviewer for his/her positive evaluation of our work. As for the last comment, we believe that this paper will be of great interest even outside the research field of PCSELS. As we explained in the introduction of the manuscript, conventional semiconductor lasers involve fundamental difficulties for single-mode high-power operation because an increase of the device size inevitably results in the onset of multiple transverse-mode lasing. Therefore, how much output power we can obtain from a single-mode semiconductor laser is an open question for all the researchers involved in the development of semiconductor lasers for decades. In addition, 100-W-to-kW-class single-mode PCSELS proposed in this paper can replace conventional bulky high-power lasers such as gas lasers, solid-state lasers and fiber lasers with a single chip. Such ultra-compact high-power semiconductor lasers will bring innovations in a wide variety of industries using lasers such as material processing, mobility, medicine, and even satellite applications. Therefore, the general recipe we propose in this study to realize 100-W-to-1-kW single-mode operation in semiconductor lasers will be of great interest not only in PCSEL communities but also for all the researchers and engineers involved in laser physics and laser industries.

In addition to the above, the theoretical analysis of non-Hermitian photonic band structures

performed in this study will also contribute to the development of non-Hermitian photonics, which is currently the subject of great academic interest. In this study, we introduce a new concept of “non-Hermitian coupling coefficients” to analytically derive the frequencies and radiation losses of Bloch modes, which enables a comprehensive analysis of complex photonic band structures near exceptional points inside non-Hermitian photonic crystals as shown in Fig. 4b (Fig. 3b in the original manuscript) in the main text. In addition, our prediction that such exceptional points are not ideal for single-mode large-area lasing due to the unstable oscillation is an interesting new finding for the community of non-Hermitian photonics, as Reviewer 2 also pointed out in the general comment. At a more fundamental level, we investigate how an open system, whose Hamiltonian is a sum of a Hermitian matrix and a non-Hermitian (specifically anti-Hermitian) matrix, exchanges energy with its environment. Accordingly, we believe that our analysis will be of mathematical interest in any field concerned with the study of wave phenomena in open systems described by this Hamiltonian.

Considering these two points, we believe that our paper has an impact on broad research communities, from those concerning the fundamental of laser physics and non-Hermitian wave physics, to those concerning applications including laser ranging and material processing. We have included the above discussions in the “DISCUSSION” section of the main text of the revised manuscript as follows;

DISCUSSION (p. 31)

...Our results provide universal guidelines towards the realization of one-chip kW-class next-generation semiconductor lasers, which are expected to replace conventional bulky high-power lasers, such as gas lasers, solid-state lasers and fiber lasers. Such ultra-compact high-power semiconductor lasers will bring innovation to a wide variety of industries using lasers, such as material processing [1,2], mobility [4], medicine [23], and even aerospace [24]. Our theoretical analysis, which considers not only Hermitian optical couplings but also non-Hermitian ones, also enables the detailed analysis of photonic bands around frequency gaps and exceptional points, which are attracting increasing attention in non-Hermitian photonics [25,26]. We believe that the theory established in this work will contribute to the development of a wide variety of research fields from fundamental laser physics and non-Hermitian wave physics in general to industrial applications.

Reply to Reviewer #2

General Comment

In this manuscript, the authors systematically analyzed the performance of double-lattice PCSEL designs, mainly on the aspect of single-mode stability. Although the overall paper is based on theoretical computation without experimental validation, the analysis is thorough and solid. They explicitly pointed out a design guideline towards the next generation of PCSELS with optimal single-mode emission power potentially up to over 100 W. The mathematical tools used by the authors, i.e. the 3D time-dependent coupled wave theory model, exhibits its superior functionality in guiding PCSEL designs, proved by its capability of predicting the IV curves, spectra and far-field patterns at varying injection currents. Furthermore, the authors' analysis on the lasing stability of the accidentally degenerate exceptional point is valuable. From the aspect of spatial hole burning, they point out that such exceptional points are not ideal as lasing candidates due to the unstable oscillation.

Although the authors didn't explicitly emphasize, this claim is different from some previous literature, such as [Opt. Lett. 39, 2072 (2014)], and will be helpful and inspiring for other groups in the community of photonic crystals and PCSELS.

However, there are some aspects in this manuscript where I think could be better optimized. I suggest the manuscript be accepted after addressing the following comments:

Reply

We are grateful to the reviewer for his/her positive evaluation of our results. We are greatly encouraged by the reviewer's comment. Each of the reviewer's comments is addressed below.

Comment 1

Generally, the presentation of the Coupled wave theory is quite mathematical in this paper. In addition to the variables defined in previous publications of the authors ($R_x, S_x, R_y, S_y, \kappa_{1D}, \kappa_{2D}$), they sequentially introduced several new mathematical terms in this paper, such as: μ (line 127), θ_{pc} (line 132), $\kappa_{1D} + \kappa_{2D}$ (line 163), R, I (lines 179, 180). However, the physical meanings of these parameters are, to me, rather abstract and obscure. For example, $\kappa_{1D} + \kappa_{2D}$ is defined as the "effective Hermitian coupling coefficient" (line 163). In some sections, the authors analyzed the model by tuning the real and the imaginary parts of this coefficient (Fig. 2). In some other sections, the analysis is then based on the "phase-invariant effective Hermitian coupling coefficient" which is essentially the previous coefficient multiplied by a phase term $\exp(-i\theta_{pc})$ (Fig. 3). Indeed, the model itself is complete and strict. However, a direct and clear expression of the physical nature of the CWT, especially in explaining the single-mode stability, might be

hindered by such a way of presentation. And the readers might find it difficult to follow the authors' framework, or the physical intuition. If possible, I suggest the authors re-express the model in a more compact way and explain the defined parameters with clear and explicit physical meanings. I believe by doing so, the potential impact of this manuscript could be enhanced. Specifically, it is not clear to me why the term $\kappa_{1D}+\kappa_{2D}$ (line 163) is a specially "effective" coefficient. The definition of θ_{pc} (line 132) is given in the Supplementary, yet the physical meaning in terms of "phase" is not clearly explained. Consequently, I don't fully understand why $(\kappa_{1D}+\kappa_{2D})\exp(-i\theta_{pc})$ is then "phase-invariant". I am confused about the necessity of defining both the "effective coefficient" and the "phase-invariant" version of this coefficient at the same time. The difference in the physical meanings of them is not clearly explained. Furthermore, I understand that the term μ can be tuned by adjusting the cladding thickness, but I'm not clear how θ_{pc} can be independently tuned by adjusting the geometrical parameters of the PCSEL.

Reply

We thank the reviewer for these insightful comments. We apologize for our confusing way of presentation of the coupled wave theory. To accurately answer the reviewer's questions, we have summarized the points of them as follows.

- 1-1. Physical meanings of "effective Hermitian coupling coefficient" ($\kappa_{1D}+\kappa_{2D}$) is not clear.**
- 1-2. Physical meanings of θ_{pc} and its relationship with the geometrical parameters of the PCSEL are not clear.**
- 1-3. The difference between "effective coefficient" ($\kappa_{1D}+\kappa_{2D}$) and "phase-invariant effective coefficient" $(\kappa_{1D}+\kappa_{2D})\exp(-i\theta_{pc})=R+iI$ is not clearly explained.**

Below, we explain each point in detail.

1-1. Physical meanings of "effective Hermitian coupling coefficient" ($\kappa_{1D}+\kappa_{2D}$)

Here, we explain the physical meaning of $\kappa_{1D}+\kappa_{2D}$. This coupling coefficient is related to the Hermitian couplings inside the photonic crystal, which do not accompany any loss. For pedagogical purposes, we first ignore the non-Hermitian couplings and focus on only the Hermitian matrix $C_{\text{Hermitian}}$. As is described in the Supplementary Section 2, the Hermitian coupling among the four fundamental waves can be expressed with $C_{\text{Hermitian}}$ as the following equation (Eq. (R1)), and can be divided into the successive equations (Eqs. (R2) and (R3)) by basis transformation considering the reflection symmetry of the photonic crystal along $y=x$:

$$(\delta + i\alpha/2) \begin{pmatrix} R_x \\ S_x \\ R_y \\ S_y \end{pmatrix} = \begin{pmatrix} \kappa_{11} & \kappa_{1D} & \kappa_{2D+} & \kappa_{2D-} \\ \kappa_{1D}^* & \kappa_{11} & \kappa_{2D-}^* & \kappa_{2D+} \\ \kappa_{2D+} & \kappa_{2D-} & \kappa_{11} & \kappa_{1D} \\ \kappa_{2D-}^* & \kappa_{2D+} & \kappa_{1D}^* & \kappa_{11} \end{pmatrix} \begin{pmatrix} R_x \\ S_x \\ R_y \\ S_y \end{pmatrix} \quad \dots(R1)$$

$$\Leftrightarrow \begin{cases} \text{For anti-symmetric electric fields} \\ (\delta + i\alpha/2) \begin{pmatrix} R_x + R_y \\ S_x + S_y \end{pmatrix} = \begin{pmatrix} \kappa_{11} + \kappa_{2D+} & \kappa_{1D} + \kappa_{2D-} \\ \kappa_{1D}^* + \kappa_{2D-}^* & \kappa_{11} + \kappa_{2D+} \end{pmatrix} \begin{pmatrix} R_x + R_y \\ S_x + S_y \end{pmatrix} \quad \dots(R2) \\ \text{For symmetric electric fields} \\ (\delta + i\alpha/2) \begin{pmatrix} R_x - R_y \\ S_x - S_y \end{pmatrix} = \begin{pmatrix} \kappa_{11} - \kappa_{2D+} & \kappa_{1D} - \kappa_{2D-} \\ \kappa_{1D}^* - \kappa_{2D-}^* & \kappa_{11} - \kappa_{2D+} \end{pmatrix} \begin{pmatrix} R_x - R_y \\ S_x - S_y \end{pmatrix} \quad \dots(R3) \end{cases}$$

The physical meanings of the above coupled-wave equations are shown in Fig. R1. Figure R1a shows the electric field vectors of the four fundamental waves in a general case, which are coupled with each other according to Eq. (R1), where \mathbf{e}_x and \mathbf{e}_y are unit vectors in the x and y directions, respectively. As shown in the above equations, the electric field vectors shown in Fig. R1a can be expressed with the superposition of those in Fig. R1b and Fig. R1c; Figure R1b corresponds to Eq. (R2), where the electric-field pairs that have anti-symmetric vectors about the axis of $y=x$ with the amplitudes of $R_x + R_y$ and $S_x + S_y$ are coupled with each other, while Fig. R1c corresponds to Eq. (R3), where the electric-field pairs that have symmetric vectors about the axis of $y=x$ with the amplitudes of $R_x - R_y$ and $S_x - S_y$ are coupled with each other. The coupling coefficients between the anti-symmetric electric-field pairs ($R_x + R_y$ and $S_x + S_y$) in Fig. R1b are given by $\kappa_{1D} + \kappa_{2D-}$ and $(\kappa_{1D} + \kappa_{2D-})^*$ as described in Eq. (R2) (see the non-diagonal terms). Therefore, $\kappa_{1D} + \kappa_{2D-}$ expresses the Hermitian coupling coefficient for the anti-symmetric electric-field pairs ($R_x + R_y$ and $S_x + S_y$). Similarly, the Hermitian coupling coefficients between the symmetric electric-field pairs ($R_x - R_y$ and $S_x - S_y$) are given by $\kappa_{1D} - \kappa_{2D-}$ and $(\kappa_{1D} - \kappa_{2D-})^*$ as described in Eq. (R3) (see the non-diagonal terms). In this paper, we define the band-edge modes generated in Fig. R1b as the anti-symmetric modes A and C, while those in Fig. R1c as the symmetric modes B and D.

In the main text (p. 9~12) of the revised manuscript, we have included Eqs. (R1)-(R3) and Fig. R1 to discuss the physical meaning of the Hermitian coupling coefficients $\kappa_{1D} + \kappa_{2D-}$ and $\kappa_{1D} - \kappa_{2D-}$. Note that in the original manuscript, we did not include Fig. R1 to explain these coefficients, and instead just used the word ‘‘effective Hermitian coupling coefficients’’ to convey the nuance of the above explanation. Now that we have included Fig. R1 to clearly explain the meaning of these coefficients, we consider that the word ‘‘effective’’ is no longer necessary. Thus, in the revised manuscripts, we have omitted the word ‘‘effective’’ and have simply called $\kappa_{1D} + \kappa_{2D-}$ and $\kappa_{1D} - \kappa_{2D-}$ the Hermitian

coupling coefficients for the anti-symmetric and symmetric modes, respectively.

Fig.R1. Physical meaning of the effective Hermitian coupling coefficients. **a**, Electric field vectors of the four fundamental waves in a general case. **b**, Electric field vector pairs that have anti-symmetric vectors about the axis of $y=x$ with the amplitudes of $R_x + R_y$ and $S_x + S_y$, where their Hermitian coupling coefficient is expressed as $\kappa_{1D} + \kappa_{2D}$ and $(\kappa_{1D} + \kappa_{2D})^*$. The resultant anti-symmetric modes are defined as modes A and C. **c**, Electric field vector pairs that have symmetric vectors about the axis of $y=x$ with the amplitudes of $R_x - R_y$ and $S_x - S_y$, where their Hermitian coupling coefficient is expressed as $\kappa_{1D} - \kappa_{2D}$ and $(\kappa_{1D} - \kappa_{2D})^*$. The resultant symmetric modes are defined as modes B and D.

1-2. Physical meanings of θ_{pc} and its relationship with the geometrical parameters of the PCSEL.

As we have shown in Fig. 1b in the main text, the fundamental waves (R_x, R_y, S_x, S_y) are coupled with each other also via a radiation wave, namely, through the non-Hermitian processes. There are two types of non-Hermitian processes: the self-coupling and the $\pm 180^\circ$ -coupling. The non-Hermitian self-coupling means that each of the fundamental waves couples to itself via a radiation wave (for example, $R_x \Rightarrow$ radiative wave $\Rightarrow R_x$), while the non-Hermitian $\pm 180^\circ$ -coupling represents the coupling between the counterpropagating fundamental waves via a radiation wave (for example, $S_x \Rightarrow$ radiative wave $\Rightarrow R_x$).

The phase change associated with the non-Hermitian $\pm 180^\circ$ -coupling process is defined as “ θ_{pc} ”, which is determined by the relative position at which non-Hermitian couplings occur when the center of the unit cell is set as the origin of the coordinate system ($x=0$). First, let us consider the case of a simple single-lattice photonic crystal whose circular air hole is placed at the center of the unit cell ($x=0$) (see Fig. R2a) as a reference. In this case, the position ($x=x_r$) at which non-Hermitian couplings occur via a radiative wave is exactly the same as $x=0$, and the phase change θ_{pc} in the non-Hermitian $\pm 180^\circ$ -coupling is equal to π because of our presupposition that the direction of the electric field vector

is opposite for R_x and S_x . On the other hand, in the case of double-lattice photonic crystals shown in Fig. R2b, the situation becomes different: when the centroids of the two air holes are equidistant from the center of the unit cell ($x=0$), the position ($x=x_r$) at which the non-Hermitian couplings occur deviates from $x=0$. This is because the position $x=x_r$ is mainly determined by the center of gravity of the refractive index distribution, which is asymmetric with respect to the origin $x=0$ in the double-lattice photonic crystal. Accordingly, the phase change θ_{pc} in the non-Hermitian $\pm 180^\circ$ -coupling becomes $2\beta x_r + \pi$ as shown in Fig. R2b, where $\beta = 2\pi/a$. In our numerical analysis, θ_{pc} is almost constant, at 0.91π , because the variance of the hole parameters (d and $2x$) are no larger than a few nanometers. We have included Fig. R2 in Supplementary Section 1 of the revised manuscript to explain the physical meaning of θ_{pc} in more detail.

Fig. R2| Physical meaning of θ_{pc} . **a**, Schematic illustration of θ_{pc} in single-lattice photonic crystal. **b**, Schematic illustration of θ_{pc} in double-lattice photonic crystal.

1-3. Difference between “effective Hermitian coupling coefficient” ($\kappa_{1D} + \kappa_{2D}$) and “phase-invariant effective Hermitian coupling coefficient” ($\kappa_{1D} + \kappa_{2D}$) $\exp(-i\theta_{pc}) = R + iI$.

$\kappa_{1D} + \kappa_{2D}$ represents the Hermitian coupling coefficient between the combined waves of $R_x + R_y$ and $S_x + S_y$ for the anti-symmetric modes as explained in the above 1-1, where we have omitted the word of “effective”. Now let us explain why we introduced “phase-invariant effective Hermitian

coupling coefficient" $(\kappa_{1D}+\kappa_{2D})\exp(-i\theta_{pc})=R+iI$. Although it is a technical issue, there is a degree of freedom in determining the phase of $\kappa_{1D}+\kappa_{2D}$. (see Fig. R3); the phase of $\kappa_{1D}+\kappa_{2D}$ was originally defined by fixing the center of double holes at the center of the unit cell ($x=0$) as described in the left panel of Fig. R3. However, the phase can change following arbitrary global translations of the air holes inside the unit cell (see the right panel of Fig. R3). To eliminate this degree of freedom, we introduced the term of $(\kappa_{1D}+\kappa_{2D})\exp(-i\theta_{pc})$ as "phase-invariant effective Hermitian coupling coefficient", where the addition of " $\exp(-i\theta_{pc})$ " allows us to consider the phase of $\kappa_{1D}+\kappa_{2D}$ relative to the phase of non-Hermitian $\pm 180^\circ$ -coupling θ_{pc} ; it physically means that the position at which the non-Hermitian $\pm 180^\circ$ -coupling occurs ($x=x_r$) is taken as a positional reference for determining the phase of the Hermitian coupling coefficients for anti-symmetric modes. In doing so, we can systematically clarify the radiation properties of the photonic crystal without mixing our results with arbitrary phase shifts induced by global translations of the holes (more details are also provided in the reply to Comment 4). We have included the above discussion on the physical meaning of " $(\kappa_{1D}+\kappa_{2D})\exp(-i\theta_{pc})$ " in the main text (p.12~13) and Supplementary Section 1 of the revised manuscript.

Finally, we would like to explain the necessity (or convenience) of defining both the coupling coefficient of " $\kappa_{1D}+\kappa_{2D}$." and the "phase-invariant" version of this coefficient " $(\kappa_{1D}+\kappa_{2D})\exp(-i\theta_{pc})=R+iI$ ". As we explained in the above paragraph, the latter ($R+iI$) contains the information about the relative phase difference between Hermitian $\pm 180^\circ$ -coupling and non-Hermitian $\pm 180^\circ$ -coupling, which enables us to systematically discuss the radiation properties of the photonic crystal. Therefore, most of the theoretical analyses in our paper were performed with the phase-invariant effective Hermitian coupling coefficient (R and I). On the other hand, the former coefficient ($\kappa_{1D}+\kappa_{2D}$), which is defined by fixing the middle point of double holes at the center of the unit cell ($x=0$), is not always necessary but useful from the viewpoint of experiments because it allows us to easily predict the change of the real and imaginary parts of $\kappa_{1D}+\kappa_{2D}$ when the structural parameters of the double-lattice photonic crystal (d and $2x$) are varied, as shown in Fig. 5d and Supplementary Section 7. Therefore, in the revised version of our manuscript, we use $\kappa_{1D}+\kappa_{2D}$ in parallel with $R+iI$ for the plot of Fig. 3 (Fig. 2 in the original manuscript) and Fig. 5d (Fig. 4d in the original manuscript); these data, when presented in this way, ought to be useful for future experimental demonstrations.

Fig. R3 | Phase change of coupling coefficients following a global translation of the air holes along $y=x$.

Comment 2

After reading the manuscript, I understood one novelty in this manuscript is the consideration of the “non-Hermitian optical couplings inside PCSELS” (line 62, lines 126-134). However, I later find the non-Hermitian coupling matrix $C_{\{\text{non-Hermitian}\}}$ (Eq. 4) is very similar to the $C_{\{\text{rad}\}}$ defined in previous publication (eg. A13 of [PRB 99, 035308 (2019)], A17 of [PRB 84, 195119 (2011)]). To avoid the potential misunderstanding of readers like me, it might be helpful for the authors to clarify the difference in the exact physics and implementation of CWT in this manuscript to the previously published ones, except for the newly defined parameters as mentioned in comment 1.

Reply

We thank the reviewer for the important comment. As the reviewer points out, in our previous papers, we have defined C_{rad} as a coupled-wave matrix which represents the mutual coupling of fundamental waves via radiative waves. However, since C_{rad} contains not only non-Hermitian couplings, which accompany emission loss, but also Hermitian couplings, which do not accompany loss, it was difficult to identify the precise parameters, specifically $i\mu$ and θ_{pc} , that govern the non-

Hermitian behavior of PCSELS. Thus, the key difference between this paper and the previous paper is that we decompose C_{rad} into its Hermitian and non-Hermitian components, which allows us to isolate (or newly define) the parameters $i\mu$ and θ_{pc} in the non-Hermitian matrix $C_{non-Hermitian}$. Physically, $i\mu$ stands for the coupling coefficient which expresses the vertical emission loss for each of the four fundamental waves during the self-coupling via a radiation wave, while θ_{pc} stands for the phase shift during the cross-coupling of counterpropagating fundamental waves via radiative waves as described in the reply to Comment 1-2. By isolating $i\mu$ and θ_{pc} , we are able to write succinct analytical expressions for the complex eigenfrequencies of each band-edge mode of the PC slab. Next, with these expressions, we are able to describe the general conditions for realizing single-mode lasing for the entire class of large-area square-lattice PC slabs.

Although we explained the relationship between $C_{non-Hermitian}$ and C_{rad} with mathematical equations in Supplementary Section 1 of our original manuscript, we did not clarify the difference in the exact physics and implementation of CWT in the main text. According to the reviewer's suggestion, we included the explanations on the difference of $C_{non-Hermitian}$ and C_{rad} as well as the purpose of the reconstruction of the coupled-wave matrices in the main text (p. 8) of the revised manuscript.

Comment 3

At lines 172 and Fig. 2, the authors choose mode C as the counterpart of target mode A. However, it is not clear from Eq. 8 that the loss of mode B is necessarily larger than that of mode C. In addition, the authors also mentioned that in the highly symmetric cases, the radiation losses of mode B is zero (Eq. 12, line 217). Therefore, I wonder what is the result if also taking mode B as the counterpart of mode A in a similar analysis.

Reply

We thank the reviewer for this comment. As we have explained in the reply to Comment 1-1, the electric field distributions of modes (A, C) and modes (B, D) have a different symmetry, where the modes A and C are derived from Eq. (R2) while the modes B and D are derived from Eq. (R3). Therefore, the reason why we choose the mode C as the counterpart of the mode A is not because the loss of mode B is larger than that of mode C, but because the modes A and C are derived from the same matrix equation [Eq. (R2)] with the same effective Hermitian coupling coefficient and have the same symmetry of electric field distributions. We have included this information in the main text (p.13~14) of the revised manuscript.

Next, we discuss the radiation loss of the mode B. When a photonic crystal has a rotational symmetry such as C_2 or C_4 , the radiation loss of mode B becomes zero as described in the manuscript. However, when the photonic crystal does not have C_2 symmetry as in the case of a double-lattice structure discussed in this work, and when $|\kappa_{1D} + \kappa_{2D}| \sim 0 \text{ cm}^{-1}$, the radiation constant of mode A can be

much lower than that of mode B, which has been numerically shown in Supplementary Fig. S2 (Fig. S1 in the original version). As shown in the figure, the radiation constant of mode A can be adjusted to less than 20 cm^{-1} via the control of $(\kappa_{1D} + \kappa_{2D}) \exp(-i\theta_{pc}) = R + iI$, while that of mode B is much higher ($>40 \text{ cm}^{-1}$) for almost all the structures. The radiation constant difference between these two modes ($>20 \text{ cm}^{-1}$) is one order of magnitude larger than the threshold margin of the fundamental mode and higher-order modes within the mode A in a large-area device shown in Fig. 4c (Fig. 3c in the original version). Therefore, it can be safely concluded that the lasing at mode B is completely suppressed by using a photonic crystal that does not have C_2 symmetry and satisfies $|\kappa_{1D} + \kappa_{2D}| \sim 0 \text{ cm}^{-1}$. We have also included this information in the main text (p.13~14) and Supplementary Section 3 of the revised manuscript.

Comment 4

At line 180, the authors assume that “ $|I|$ is much smaller than $|R+i\mu|$ ”. Could they explain the physical meaning of such an assumption?

Reply

We thank the reviewer for the important question. We first review the role of the non-Hermitian coupling coefficient ($i\mu$) in radiation process. Physically, $i\mu$ represents the vertical emission loss that accompanies each of the four fundamental waves as it couples to itself via radiative waves. For example, the radiation constant of each band-edge mode (A, B, C, D) becomes almost equal to 2μ when their in-plane k -vectors ($k_{//}$) deviate far enough from the Γ point, where the effect of mutual couplings among the four fundamental waves becomes very weak.

On the other hand, at the Γ point of each band, the mutual couplings among the four fundamental waves determine the overall behavior of radiation from the four fundamental waves. As we have explained in the reply to Comment 1-1, the Hermitian couplings between the two anti-symmetric electric-field pairs ($R_x + R_y$ and $S_x + S_y$) in modes A and C are determined by $[\kappa_{1D} + \kappa_{2D}, (\kappa_{1D} + \kappa_{2D})^*]$, and there are also non-Hermitian $\pm 180^\circ$ -couplings $[i\mu \exp(\pm i\theta_{pc})]$ between these pairs. Combining these two effects, the coupling matrix for the modes A and C is derived as follows (a detailed derivation is provided in Supplementary Section 2):

$$\begin{aligned}
 (\delta + i\alpha/2) \begin{pmatrix} R_x + R_y \\ S_x + S_y \end{pmatrix} &= \begin{pmatrix} \kappa_{11} + \kappa_{2D+} + i\mu & \kappa_{1D} + \kappa_{2D-} + i\mu e^{i\theta_{pc}} \\ \kappa_{1D}^* + \kappa_{2D-}^* + i\mu e^{-i\theta_{pc}} & \kappa_{11} + \kappa_{2D+} + i\mu \end{pmatrix} \begin{pmatrix} R_x + R_y \\ S_x + S_y \end{pmatrix} \\
 &= \begin{pmatrix} \kappa_{11} + \kappa_{2D+} + i\mu & [R + iI + i\mu] e^{i\theta_{pc}} \\ [R - iI + i\mu] e^{-i\theta_{pc}} & \kappa_{11} + \kappa_{2D+} + i\mu \end{pmatrix} \begin{pmatrix} R_x + R_y \\ S_x + S_y \end{pmatrix} \quad \dots \quad (\text{R4})
 \end{aligned}$$

From this equation, one can easily understand that the coupling strengths from $S_x + S_y$ to $R_x + R_y$ and vice versa are different when the imaginary part of the phase-invariant effective Hermitian coupling

coefficients (I) takes a non-zero value. This asymmetry in the coupling strengths determines the overall radiation constant of each band-edge mode (see the next paragraph for further details). Specifically, when the condition of $|I| \ll |R+i\mu|$ is satisfied, the coupling strengths from $S_x + S_y$ to $R_x + R_y$ and vice versa are almost equal, which leads to a relatively small radiation constant of mode A ($<20\sim30\text{cm}^{-1}$) (as also described in the next paragraph). This enables lasing with a low-to-moderate threshold current, which is desirable in a practical laser. This information has been added to Supplementary Section 4 of the revised manuscript.

To visually understand the physical meaning of the imaginary part of the effective Hermitian coupling coefficient (I), we calculate the electric field distributions of mode A for typical double-lattice photonic crystals with different values of I (corresponding to different values of $2x$ in Fig. 5b). The results of these calculations are shown in Fig. R4, where mode A has a node (shown in blue) around which the electric field vectors circulate. The position of this node is determined by the above Eq. (R4), where the vertical radiation is cancelled out. When the value of $|I|$ is almost zero (middle panel), the position of this node coincides with the position of the non-Hermitian coupling (x_r) explained in Fig. R2 (shown in red), leading to complete cancellation of the radiation for mode A. On the other hand, when the value of $|I|$ is larger than zero (left and right panels), the position of the node deviates from x_r , leading to the incomplete cancellation of the radiation and the increase of the radiation constant. Thus, the imaginary part of the phase-invariant effective Hermitian coupling coefficient (I) determines the degree of cancellation of the vertical radiation at the Γ point, which is based on the deviation of the node of the electric field from the position of non-Hermitian couplings. We have included the above discussions and Fig. R4 in Supplementary Section 4 of the revised manuscript.

Fig. R4| Electric field distributions of mode A in double-lattice photonic crystals with different values of I . Arrows indicate the electric field vectors. The red dot indicates the position for non-Hermitian couplings, while the blue dot indicates the node of the electric field distribution.

Comment 5

At line 205, the authors mention that modes A and C are degenerate at “ Γ point”. I wonder what is the definition of the “ Γ point” here - is it the point of $R=I=0$ in Fig. 2, or the generally accepted definition of $k=0$ point in a band structure? If the latter, I think it helps to show the band structure somewhere (maybe in the Supplementary). If the former, please clarify it.

Reply

We thank the reviewer for the comment. Here, the meanings of “ Γ point” is the same as the generally accepted definition of $k=0$ point in a band structure. The band structure with the degeneracy of modes A and C when $R=0$ is already shown in Fig. 4b (Fig. 3b in the original manuscript) with a red line.

Comment 6

I’m confused at the figure of merit in optimising the single-mode stability. In Fig. 3b and lines 237-249, the authors are emphasising on the “sensitivity of change in α_A and δ_{AC} with respect to Δk . However, why not the figure of merit as “the maximum difference in the loss for all Δk ”? After all, this is closer to the difference in lasing threshold between the modes, which is related to the single-mode stability in a more direct way. As shown in the right panel of Fig. 3b, the case of $R=160\text{cm}^{-1}$ allows the maximum difference between α_A and α_C for all Δk , yet this is the case which is the least steep.

Reply

We thank the reviewer for the comment. As we have explained in the reply to Comment 3, when we consider a photonic crystal that completely breaks C_2 symmetry and also satisfies $|\kappa_{1D}+\kappa_{2D}| \sim 0 \text{ cm}^{-1}$, the radiation constant of mode A can be selectively reduced with respect to those of the other three modes (B, C, D), and the radiation constant difference between mode A and the others can be as large as $>20\text{cm}^{-1}$. On the other hand, when we increase the device size beyond 3 mm, the difference of the in-plane wavenumber of the fundamental mode ($\Delta k \sim \pi/L$) and the first higher order modes ($\Delta k \sim 2\pi/L$) within a single band becomes very small, which leads to a much smaller radiation constant difference $\Delta\alpha_v$ between the fundamental mode and the higher order modes within that band, as shown in Fig. 4c (Fig. 3c in the original manuscript). **Therefore, it is not the “inter-band” radiation constant difference but the “intra-band” radiation constant difference that determines the single-mode stability of large-area PCSELS, and the sensitivity of change in α_A with respect to Δk (or $d\alpha_A/d\Delta k$) is the most important figure of merit for determining this intra-band difference.**

For example, in the case of $R=160\text{ cm}^{-1}$, the radiation constant difference between mode A and mode C is large, but the sensitivity of α_A to changes of Δk is small; thus, the case of $R=160\text{ cm}^{-1}$ is not suitable for single-mode lasing. In the main text (p.17) of the revised manuscript, we explain the reason why we focus on the sensitivity of the change of α_A with respect to Δk in more detail.

Comment 7

In pages 13-15, the authors considered the cases with Δk only in Γ -M and Γ -M' directions. What about Γ -X directions, or other arbitrary directions in the vicinity of the Γ -point? The choice of selection here should be better justified that it would not lose the generality.

Reply

We thank the reviewer for this important comment. In our study, we consider a photonic crystal which has a reflection symmetry along $y=x$ (Γ -M direction), and we assume that the shape of the current injection region is circular, which also maintains the reflection symmetry along $y=x$. As a result, the electric field distribution of the eigenmodes in the finite-sized PCSEL becomes either symmetric or anti-symmetric along $y=x$, which always yields higher-order modes with two antinodes in the Γ -M and Γ -M' directions, respectively. For example, Fig. R5 shows the calculated threshold margins of all the eigenmodes in the 3-mm-diameter double-lattice PCSEL, which was designed for the simulation in Fig. 6 ($R=45\text{ cm}^{-1}$, $\mu=44\text{ cm}^{-1}$, $I=32\text{ cm}^{-1}$), as well as the electric-field intensity distributions of the five lowest-threshold modes of band-edge A. As seen in the figure, the high-order modes that have the lowest and second lowest threshold gain (A-2 and A-3) have two antinodes in the Γ -M and Γ -M' directions, so their radiation constants can be described by simply considering Δk along the Γ -M and Γ -M' directions. Other higher-order modes (A-4, A-5) have more complex electric field distributions, but their threshold gains are much larger. Therefore, in the discussion of the single-mode stability of PCSELS with reflection symmetry along $y=x$, it is sufficient to consider the wavenumber dependence of the radiation constants in the Γ -M direction and the Γ -M' direction as discussed in the main text. We have included Fig. R5 as Supplementary Fig. S4 in Supplementary Section 5 of the revised manuscript, which is also related to Comment 8.

Fig. R5| Calculated total loss and electric field distributions of eigenmodes in a finite-sized PCSEL ($L=3$ mm, $R=45$ cm^{-1} , $\mu=44$ cm^{-1} , $I=32$ cm^{-1}).

Comment 8

The “threshold margin” $\Delta\alpha_v$ (Eq. 15, Fig. 3c) only considered the radiation loss difference between the fundamental mode and its own high order mode, without considering other fundamental modes. To me, this is not fully precise and strict. Firstly, although for square-lattice PC, there are only 4 modes in the infinite case near Γ_2 point, however there are 6 fundamental modes in the finite case, as mentioned by an early publication of the authors’ group [Sec. 3.2, Opt. Express 15, 3981-3990 (2007)], where fundamental modes O and O’ are seemingly exclusive to finite structures. Secondly, as shown in Fig. 2i of a prior publication from the authors’ group [Opt. Express 20, 15945(2012)], although the lowest loss mode there is the fundamental mode A, the one with the second lowest loss is not its first high-order mode, but modes from other groups. This is also the case in other publications, such as Fig. 2b of [Appl. Phys. Lett. 104, 021102 (2014)], Fig. 3a of [Opt. Express 15, 3981-3990 (2007)]. Therefore, I think it is more convincing to show a mode spectrum ($\alpha_{\text{total}} L$ vs δL) as these mentioned figures in prior publications, where the overall cavity losses of all finite fundamental modes and their high order modes can be clearly seen. And the authors could define another “global threshold margin” as the difference in threshold between the lowest two modes in such a figure, then analyse its robustness. Such a “global” analysis would be highly convincing and helpful in support to the conclusions of this manuscript.

Reply

We thank the reviewer for this important comment. If the reviewer would allow us to jump straight to the conclusion, our analysis is precise and strict for large-area (> 3 mm) devices. As we have already explained in the replies to Comment 3 and 6, when we consider a photonic crystal that completely breaks C_2 symmetry and also satisfies $|\kappa_{1D} + \kappa_{2D}| \sim 0$ cm^{-1} , the radiation constant of mode A can be selectively reduced compared to those of the other three modes (B, C, D), and the radiation constant difference between mode A and the others is as large as >20 cm^{-1} . On the other hand, the radiation constant difference between the fundamental mode and higher order modes of a 3-mm-diameter PCSEL is much smaller (<5 cm^{-1}) as shown in Fig. 4c (Fig. 3c in the original manuscript). Therefore, in a case of a large-area (3~10 mm) PCSEL with $|\kappa_{1D} + \kappa_{2D}| \sim 0$ cm^{-1} that completely breaks C_2 symmetry, we can ignore the mode competition between different band-edge modes, unlike small PCSELS considered in the previous literatures [Opt. Express **15**, 3981-3990 (2007), Opt. Express **20**, 15945(2012), Appl. Phys. Lett. **104**, 021102 (2014)]. To clarify this point, we calculated the mode spectrum of the 3-mm-diameter double-lattice PCSEL, which was designed for the simulation in Fig. 6 ($R=45$ cm^{-1} , $\mu=44$ cm^{-1} , $I=32$ cm^{-1}), according to the suggestion of the reviewer. The result is already shown in Fig. R5 in the reply to Comment 7. From the figure, it is obvious that the eigenmodes originating from the band-edges B, C, and D have much higher losses than the eigenmodes originating from band-edge A. It should be also noted that mode O and O' shown in the previous paper [Opt. Express **15**, 3981-3990 (2007)] appear only when the magnitude of the non-Hermitian coupling constant μ (which is proportional to the average radiation constant of the four band-edge modes) is small [note that dimensionless non-Hermitian coupling constant in this paper ($\mu L = 44$ $\text{cm}^{-1} \times 3$ mm = 13.2) is two orders of magnitude larger than that assumed in the above paper ($\kappa_0 L = 0.09$), where TM-polarized modes were considered]. **Therefore, it can be concluded that the threshold margin between the fundamental mode and higher order modes of band-edge A ($\Delta\alpha_v$) is also the global threshold margin of the large-area PCSELS.** We have included the above discussion and Fig. R5 in Supplementary Section 5 of the revised manuscript.

Comment 9

As essentially a proposal, the authors could also discuss the potential challenges in experimentally realizing the 100W or 1kW PCSELS proposed in this manuscript, if there is any. For example, would the current injection, or the thermal dissipation be a problem? If yes, how can they be overcome? Are there any other technical difficulties?

Reply

We thank the reviewer for this important comment. As the reviewer has pointed out, it is important to control the current density distribution of the device as well as to dissipate a large amount of heat generated inside of the device. Concerning the former, it is difficult to inject current into the center of

a large-area device using a conventional ring-window electrode; to realize a spatially uniform current distribution, a mesh-type electrode, as was investigated in Ref. R1 for example, must be employed instead. In the case of a 3-mm-diameter PCSEL, by employing the mesh-type electrode shown in Fig. R6a, we can realize the current density distribution shown in Fig. R6b, in which the current density at the center of the electrode is only 20% lower than that at the edge. This current distribution is almost equal to that assumed in our time-domain simulations. It should be noted that we have numerically confirmed that the impact of the mesh on the beam profile is small because the width of each mesh (15 μm) is two orders of magnitude smaller than the diameter of the device (3 mm). In addition, the employment of the mesh-type electrode also contributes to the reduction of the series resistance of the device, which is useful to decrease the heat generated inside the device, which is discussed in the next paragraph. A similarly designed mesh-type electrode can be also applied to 10-mm-diameter PCSELS.

As for thermal management, the amount of heat generated inside of the device is negligible in the case of short-pulse operation with pulse widths of less than 1 μs , which is used for LiDAR and micro-processing. For continuous-wave (CW) operation, however, it is necessary to mount the device to a cooling package for heat dissipation. In this case, since the maximum heat dissipation per unit area is fixed, the maximum CW emission power is nearly proportional to the device area. Based on a CW emission power of $\sim 7\text{W}$ achieved using an 800- μm -diameter PCSEL [R2], it is expected that CW emission powers of $\sim 98\text{ W}$ and $\sim 1090\text{ W}$ are achievable using 3-mm-diameter and 10-mm-diameter PCSELS, respectively. For example, we have calculated that, for a 3-mm-diameter PCSEL mounted to a water-cooling package at 20°C as shown in Fig. R6c, we can suppress the maximum temperature of the device to below 60°C even under a heat generation of 200 W (which corresponds to an injection current of $\sim 130\text{ A}$ and a CW output power of $\sim 100\text{ W}$) as shown in Fig. R6d. It should be also noted that the non-uniform temperature distribution shown in Fig. R6d and the resultant band-edge frequency distribution affect the lasing characteristics of the device, but the effects of these distributions can be compensated by spatially varying the lattice constant of the photonic-crystal structure [R3].

In summary, we believe that 100-W-to-1-kW-class PCSELS proposed in this paper are experimentally feasible. Development of these large-area devices is currently underway. We have included the above discussion on the potential challenges and solutions for experimentally realizing the 100-W-to-1-kW PCSELS in Supplementary Section 10 of the revised manuscript.

[R1] Z. Wang *et al*, “Large area photonic crystal quantum cascade laser with 5 W surface-emitting power,” *Opt. Express* **27**, 22708 (2019).

[R2] M. De Zoysa *et al*, “Thermal management for CW operation of large-area double-lattice photonic-crystal lasers,” *J. Opt. Soc. Am. B* **37**, 3882 (2020).

[R3] S. Katsuno *et al*, “Self-consistent analysis of photonic-crystal surface-emitting lasers under

continuous-wave operation,” Opt. Express **29**, 25118 (2021).

Fig. R6| Control of current density distribution and temperature distribution in a large-area PCSEL. a, Schematic of a mesh-type electrode on a large-area PCSEL. **b**, Calculated current density distribution of the device shown in **a**. **c**, Schematic of mounting of a large-area PCSEL to a sub-mount and a heat sink. **d**, Calculated temperature distribution of the device shown in **c**.

Comment 10

I think it helps in Fig. 5b to label also the relative relationship between R and μ in each panel. For example, “ $R \sim \mu$ ” could be labeled to the second panel.

Reply

We thank the reviewer for this helpful comment. As the reviewer has suggested, we have labelled the relative relationship between R and μ in each panel of Fig. 6b (Fig. 5b in the original manuscript).

Comment 11

I feel that there might be a typo in Fig. 1a - is a star sign (*) missing for one of the two κ_{2D+} terms?

Reply

We thank the reviewer for this comment. Actually, this is not a typo; our photonic crystal structure has the reflection symmetry along $y=x$, so the coupling from R_y to R_x and from R_x to R_y are equivalent. Thus, κ_{2D+} is equal to κ_{2D+}^* (in other words, κ_{2D+} is a real number). The above discussion is already included in the main text of the manuscript (p. 7).

Comment 12

Maybe the authors could briefly mention the potential impact of this manuscript to other groups outside the PCSEL community somewhere in the manuscript or in the Supplementary.

Reply

We thank the reviewer for this important comment. We believe that this paper will be of great interest to researchers even outside the research field of PCSELS. As we have explained in the introduction of the manuscript, conventional semiconductor lasers involve fundamental difficulties for single-mode high-power operation because an increase of the device size inevitably results in the onset of multiple transverse-mode lasing. Therefore, how much output power we can obtain from a single-mode semiconductor laser is an open question for all the researchers involved in the development of semiconductor lasers for decades. In addition, 100-W-to-kW-class single-mode PCSELS proposed in this paper can replace conventional bulky high-power lasers such as gas lasers, solid-state lasers and fiber lasers with a single chip. Such ultra-compact high-power semiconductor lasers will bring innovations in a wide variety of industries using lasers such as material processing, mobility, medicine, and even satellite applications. Therefore, the general recipe we propose in this study to realize 100-W-to-1-kW single-mode operation in semiconductor lasers will be of great interest not only in PCSEL communities but also for all the researchers and engineers involved in laser physics and laser industries.

In addition to the above, the theoretical analysis of non-Hermitian photonic band structures performed in this study will also contribute to the development of non-Hermitian photonics, which is currently the subject of great academic interest. In this study, we introduce a new concept of “non-Hermitian coupling coefficients” to analytically derive the frequencies and radiation losses of Bloch modes, which enables a comprehensive analysis of complex photonic band structures near exceptional points inside non-Hermitian photonic crystals as shown in Fig. 4b (Fig. 3b in the original manuscript) in the main text. In addition, our prediction that such exceptional points are not ideal for single-mode large-area lasing due to the unstable oscillation is an interesting new finding for the community of non-Hermitian photonics, as the reviewer also pointed out in the general comment. At a more fundamental level, we investigate how an open system, whose Hamiltonian is a sum of a Hermitian matrix and a non-Hermitian (specifically anti-Hermitian) matrix, exchanges energy with its environment. Accordingly, we believe that our analysis will be of mathematical interest in any field

concerned with the study of wave phenomena in open systems described by this Hamiltonian.

Considering these two points, we believe that our paper has an impact on broad research communities, from those concerning the fundamental of laser physics and non-Hermitian physics, to those concerning applications including laser ranging and material processing. We have included the above discussions in the “DISCUSSION” section of the main text of the revised manuscript as follows;

DISCUSSION (p. 31)

...Our results provide universal guidelines towards the realization of one-chip kW-class next-generation semiconductor lasers, which are expected to replace conventional bulky high-power lasers, such as gas lasers, solid-state lasers and fiber lasers. Such ultra-compact high-power semiconductor lasers will bring innovation to a wide variety of industries using lasers, such as material processing [1,2], mobility [4], medicine [23], and even aerospace [24]. Our theoretical analysis, which considers not only Hermitian optical couplings but also non-Hermitian ones, also enables the detailed analysis of photonic bands around frequency gaps and exceptional points, which are attracting increasing attention in non-Hermitian photonics [25,26]. We believe that the theory established in this work will contribute to the development of a wide variety of research fields from fundamental laser physics and non-Hermitian wave physics in general to industrial applications.

REVIEWERS' COMMENTS

Reviewer #2 (Remarks to the Author):

All my questions and concerns are answered clearly. The quality of the manuscript has been significantly improved. I suggest the paper be accepted.